# Reinforcement Learning with Adaptive Regularization for Safe Control of Critical Systems

**Haozhe Tian**[*]   **Homayoun Hamedmoghadam**   **Robert Shorten**   **Pietro Ferraro**
Dyson School of Design Engineering
Imperial College London
SW7 2AZ, London, UK
{haozhe.tian21, h.hamed, r.shorten, p.ferraro}@imperial.ac.uk

## Abstract

Reinforcement Learning (RL) is a powerful method for controlling dynamic systems, but its learning mechanism can lead to unpredictable actions that undermine the safety of critical systems. Here, we propose RL with Adaptive Regularization (RL-AR), an algorithm that enables safe RL exploration by combining the RL policy with a policy regularizer that hard-codes the safety constraints. RL-AR performs policy combination via a "focus module," which determines the appropriate combination depending on the state—relying more on the safe policy regularizer for less-exploited states while allowing unbiased convergence for well-exploited states. In a series of critical control applications, we demonstrate that RL-AR not only ensures safety during training but also achieves a return competitive with the standards of model-free RL that disregards safety.

## 1   Introduction

A wide array of control applications, ranging from medical to engineering, fundamentally deals with *critical systems*, i.e., systems of vital importance where the control actions have to guarantee no harm to the system functionality. Examples include managing nuclear fusion [Degrave et al., 2022], performing robotic surgeries [Datta et al., 2021], and devising patient treatment strategies [Komorowski et al., 2018]. Due to the critical nature of these systems, the optimal control policy must be explored while ensuring the safety and reliability of the control algorithm.

Reinforcement Learning (RL) aims to identify the optimal policy by learning from an agent's interactions with the controlled environment. RL has been widely used to control complex systems [Silver et al., 2016, Ouyang et al., 2022]; however, the learning of an RL agent involves trial and error, which can violate safety constraints in critical system applications [Henderson et al., 2018, Recht, 2019, Cheng et al., 2019b]. To date, developing reliable and efficient RL-based algorithms for real-world "single-life" applications, where the control must avoid unsafety from the first trial [Chen et al., 2022], remains a challenge. The existing safe RL algorithms either fail to ensure safety during the training phase [Achiam et al., 2017, Yu et al., 2022a] or require significant computational overhead for action verification [Cheng et al., 2019a, Anderson et al., 2020]. As a result, classic control methods are often favored in critical applications, even though their performance heavily relies on the existence of an accurate model of the environment.

Here, we address the safety issue of RL in scenarios where "estimated" environment models are available (or can be built) to derive sub-optimal control policy priors. These scenarios are representative of many real-world critical applications [Hovorka et al., 2002, Liepe et al., 2014, Hippisley-Cox et al., 2017, Rathi et al., 2021]. Consider the example of devising a control policy that prescribes the

---

[*]Corresponding author

optimal drug dosages for regulating a patient's health status. This is a single-life setting where no harm to the patient is tolerated during policy exploration. From available records of other patients, an estimated patient model can be built to predict the response to different drug dosages and ensure adherence to the safety bounds (set based on clinical knowledge). However, a new patient's response can deviate from the estimated model, which poses a significant challenge in control adaptability and patient treatment performance.

We propose a method, *RL with Adaptive Regularization* (RL-AR), that simultaneously shows the safety and adaptability properties required for critical single-life applications. The method interacts with the actual environment using two parallel agents. The first (safety regularizer) agent avoids unsafe states by leveraging the forecasting ability of the estimated model. The second (adaptive) agent is a model-free RL agent that promotes adaptability by learning from actual environment interactions. Our method introduces a "focus module" that performs state-dependent combinations of the two agents' policies. This approach allows immediate safe deployment in the environment by initially prioritizing the safety regularizer across all states. The focus module gradually learns to apply appropriate policy combinations depending on the state—relying more on the safety regularizer for less-exploited states while allowing unbiased convergence for well-exploited states.

We analytically demonstrate that: i) RL-AR regulates the harmful effects of overestimated RL policies, and ii) the learning of the state-dependent focus module does not prevent convergence to the optimal RL policy. We simulate a series of safety-critical environments with practically obtainable sub-optimal estimated models (e.g., from real-life sampled measurements). Our empirical results show that even with more than 60% parameter mismatches between the actual environment model and the estimated model, RL-AR ensures safety during training while converging to the control performance standard of model-free RL approaches that prioritize return over safety.

## 2 Preliminaries

Through environment interactions, an RL agent learns a policy that maximizes the expected cumulative future reward, i.e. the expected return. We formalize the environment as a Markov Decision Process (MDP) $\mathcal{M} = (\mathcal{S}, \mathcal{A}, P, r, \gamma)$, where $\mathcal{S}$ is a finite set of states, $\mathcal{A} = \{a \in \mathbb{R}^k : \underline{a} \leq a \leq \overline{a}\}$ is a convex action-space, $P : \mathcal{S} \times \mathcal{A} \to \mathcal{P}(\mathcal{S})$ is the state transition function, $r : \mathcal{S} \times \mathcal{A} \to [-R_{\max}, R_{\max}]$ is the reward function, and $\gamma \in (0, 1)$ is a discount factor. Let $\pi$ denote a stochastic policy $\pi : \mathcal{S} \to \mathcal{P}(\mathcal{A})$, the value function $V^\pi$ and the action-value function $Q^\pi$ are:

$$V^\pi(s_t) = \mathbb{E}_{a_t, s_{t+1}, \dots} \left[ \sum_{i=0}^\infty \gamma^i r(s_{t+i}, a_{t+i}) \right], \quad Q^\pi(s_t, a_t) = \mathbb{E}_{s_{t+1}, \dots} \left[ \sum_{i=0}^\infty \gamma^i r(s_{t+i}, a_{t+i}) \right], \quad (1)$$

where $a_t \sim \pi(s_t)$, $s_{t+1} \sim P(s_t, a_t)$ for $t \geq 0$. The optimal policy $\pi^\star = \arg\max_\pi V^\pi(s)$ maximizes the expected return for any state $s$. Both $V^\pi$ and $Q^\pi$ satisfy the Bellman equation [Bellman, 1966]:

$$V^\pi(s) = \mathbb{E}_{a,s'} \left[ r(s, a) + \gamma V^\pi(s') \right], \quad Q^\pi(s, a) = \mathbb{E}_{s'} \left[ r(s, a) + \gamma \mathbb{E}_{a' \sim \pi(s')} [Q^\pi(s', a')] \right]. \quad (2)$$

For practical applications with complex $\mathcal{S}$ and $\mathcal{A}$, $Q^\pi$ and $\pi$ are approximated with neural networks $Q_\phi$ and $\pi_\theta$ with learnable parameters $\phi$ and $\theta$. To stabilize the training of $Q_\phi$ and $\pi_\theta$, they are updated using samples $\mathcal{B}$ from a Replay Buffer $\mathcal{D}$ [Mnih et al., 2013], which stores each previous environment transitions $e = (s, a, s', r, d)$, where $d$ equals 1 for terminal states and 0 otherwise.

In this work, we are interested in acting on a safe regularized RL policy that can differ from the raw RL policy. RL approaches that allow learning from a different acting policy are referred to as "off-policy" RL. The RL agent in our proposed algorithm follows the state-of-the-art off-policy RL algorithm: Soft Actor-Critic (SAC) [Haarnoja et al., 2018], which uses a multivariate Gaussian policy to explore environmental uncertainties and prevent getting stuck in sub-optimal policies. For $Q$-network updates, SAC mitigates the overestimation bias by using the clipped double $Q$-learning, which updates the two $Q$-networks $Q_{\phi_i}, i = 1, 2$ using gradient descent with the gradient:

$$\nabla_{\phi_i} \frac{1}{|\mathcal{B}|} \sum_{(s,a,s',r,d) \in \mathcal{B}} (Q_{\phi_i}(s, a) - y)^2, \quad i = 1, 2,$$

$$y = r + \gamma(1 - d) \left( \min_{i=1,2} Q_{\phi_{targ,i}}(s', a') - \alpha \log P_{\pi_\theta}(a' \mid s') \right), \quad a' \sim \pi_\theta(s'), \quad (3)$$

where the entropy regularization term $\log P_{\pi_\theta}(a' \mid s')$ encourages exploration, thus avoiding local optima. Target $Q$-networks $\phi_{targ,i}$ are used to reduce drastic changes in value estimates and stabilize training. The target $Q$-network parameters are initialized with $\phi_{targ,i} = \phi_i$, $i = 1, 2$. Each time $\phi_1$ and $\phi_2$ are updated, $\phi_{targ,1}, \phi_{targ,2}$ slowly track the update using $\tau \in (0, 1)$:

$$\phi_{targ,i} = \tau \phi_{targ,i} + (1 - \tau)\phi_i, \;\; i = 1, 2. \tag{4}$$

For policy updates, the policy network $\pi_\theta$ is updated using gradient ascent with the gradient:

$$\nabla_\theta \frac{1}{|\mathcal{B}|} \sum_{s \in \mathcal{B}} \left( \min_{i=1,2} Q_{\phi,i}(s, a_\theta(s)) - \alpha \log P_{\pi_\theta}(a \mid s) \right), \quad a_\theta(s) \sim \pi_\theta(s). \tag{5}$$

## 3 Methodology

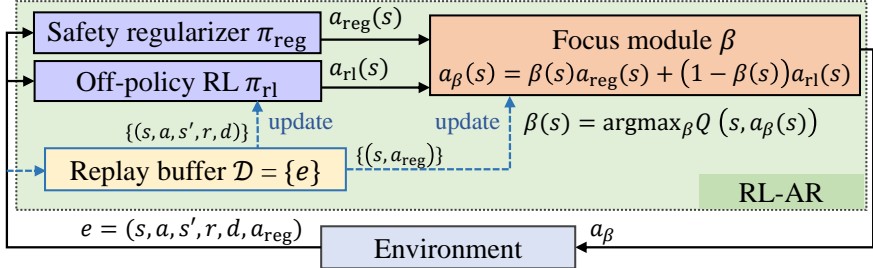

Figure 1: Schematic overview of the proposed RL-AR algorithm. RL-AR integrates the policies of the RL agent and the safety regularizer agent using a state-dependent focus module, which is updated to maximize the expected return of the combined policy.

Here, we propose RL-AR, an algorithm for the safe training and deployment of RL in safety-critical applications. A schematic view of the RL-AR procedures is shown in Fig. 1. RL-AR comprises two parallel agents and a focus module: (i) The *safety regularizer* agent follows a deterministic policy $\pi_{\text{reg}} : \mathcal{S} \to \mathcal{A}$ proposed by a constrained model predictive controller (MPC); (ii) The *off-policy RL* agent is an adaptive agent with $\pi_{\text{rl}} : \mathcal{S} \to \mathcal{P}(\mathcal{A})$ that can learn from an acting policy that is different from $\pi_{\text{rl}}$; (iii) The *focus module* learns a state-dependent weight $\beta : \mathcal{S} \to [0, 1]$ for combining the deterministic $a_{\text{reg}}(s) = \pi_{\text{reg}}(s)$ and the stochastic $a_{\text{rl}}(s) \sim \pi_{\text{rl}}(s)$. Among the components, the safety regularizer has a built-in estimated environment model $\tilde{f} : \mathcal{S} \times \mathcal{A} \to \mathcal{S}$ that is different from the actual environment model, while the off-policy RL agent and focus module are dynamically updated using observed interactions in the actual environment.

The RL-AR workflow is as follows: (i) $\pi_{\text{reg}}(s)$ generates $a_{\text{reg}}(s)$, which hard-codes safety constraints in the optimization problem over a period forecasted by $\tilde{f}$. The forecasting ability anticipates and prevents running into unsafe states for the critical system; (ii) $\pi_{\text{rl}}(s)$ generates $a_{\text{rl}}(s)$ to allow stochastic exploration and adaptation to the actual environment; (iii) $\beta(s)$ is initialized to $\beta(s) \geq 1 - \epsilon, \forall s \in \mathcal{S}$, hence prioritizing the safe $\pi_{\text{reg}}$ before $\pi_{\text{rl}}$ learns a viable policy. As more interactions are observed for a state $s$ and the expected return of $\pi_{\text{rl}}(s)$ improves, $\beta(s)$ gradually shifts the focus from the initially suboptimal $\pi_{\text{reg}}(s)$ to $\pi_{\text{rl}}(s)$.

### 3.1 The safety regularizer

The safety regularizer of RL-AR is a constrained MPC, which, at any state $s_t$, optimizes the $N$-step system behavior forecasted using the estimated environment model $\tilde{f}$ by solving the following constrained optimization problem:

$$\min_{a_{t:t+N-1}} \sum_{k=t}^{t+N-1} J_k(s_k, a_k) + J_N(s_{t+N}) \tag{6}$$

$$\text{s.t.} \quad s_{k+1} = \tilde{f}(s_k, a_k), g(s_k) \geq 0, a_k \in \mathcal{A},$$

where $J_k(s_k, a_k)$ and $J_N(s_{t+N})$ are the stage and terminal cost functions and $g(s_k) \geq 0$ is the safety constraint. By hard-coding the safety constraints in the optimization (via $g(s_k) \geq 0$) over the

---

**Algorithm 1** RL-AR

---

1: **Initialization:** empty replay buffer $\mathcal{D}$; MPC controller with estimated environment model $\tilde{f}$; policy network $\pi_\theta$; $Q$-networks $Q_{\phi_i}$ and target $Q$-networks $Q_{\phi_{targ,i}}$ with $\phi_{targ,i} = \phi_i, i = 1, 2$; pretrained focus module $\beta_\psi$ with $\beta_\psi(s) \geq 1 - \epsilon, \forall s \in \mathcal{S}$; time step $t = 0$.
2: **repeat**
3:     Observe state $s$
4:     Take action $a = \beta_\psi(s)a_{\text{reg}}(s) + (1 - \beta_\psi(s))a_{\text{rl}}(s), \quad a_{\text{reg}}(s) = \pi_{\text{reg}}(s), a_{\text{rl}}(s) \sim \pi_\theta(s)$
5:     Store transition $e = (s, a, s', r, d, a_{\text{reg}})$ in $\mathcal{D}$
6:     **if** $t > time\ to\ update$ **then**
7:         Randomly sample a batch $\mathcal{B}$ of transitions from $\mathcal{D}$
8:         Update $Q_{\phi_i}, i = 1, 2$, by gradient descent with Eq. (3)
9:         Update $\pi_\theta$ by gradient ascent with Eq. (5)
10:        Update $\beta_\psi$ by gradient ascent with:

$$\nabla_\psi \frac{1}{|\mathcal{B}|} \sum_{(s, a_{\text{reg}}) \in \mathcal{B}} \min_{i=1,2} Q_{\phi_i}\left(s, \beta_\psi(s)a_{\text{reg}} + (1 - \beta_\psi(s))a_{\text{rl}}(s)\right), \ \ a_{\text{rl}}(s) \sim \pi_\theta(s)$$

11:        Update $Q_{\phi_{targ,i}}, i = 1, 2$, using Eq. (4)
12:     **end if**
13:     $t = t + 1$
14: **until** $convergence$ is $true$

---

prediction horizon, MPC prevents failure events that are not tolerated in critical applications. MPC iteratively solves for the N-step optimal actions in each time step and steers the environment to the desired state. At any time step $t$, solving the optimization problem in Eq. (6) yields a sequence of $N$ actions $a_{t:t+N-1}$, with only the first action $a_t$ in the sequence adopted for the current time step, i.e., $a_{\text{reg}}(s_t) = \pi_{\text{reg}}(s_t) = a_t$. The system transitions from $s_t$ to $s_{t+1}$ by taking the action $a_{\text{reg}}(s_t)$, and the optimization problem is solved again over $\{t + 1 : t + 1 + N\}$ to obtain $a_{\text{reg}}(s_{t+1})$. For practical applications with continuous state space, the optimization problem in Eq. (6) is efficiently solved using the Interior Point Optimizer [Andersson et al., 2019]. Since MPC solves similar problems with slight variations at each time step, the computational complexity is further reduced by using the solution from the previous step as the initial guess.

### 3.2 Policy regularization

The focus module in RL-AR combines the actions proposed by the safety regularizer agent and the RL agent using a weighted sum. At state $s$, the combined policy $\pi_\beta$ takes the following action $a_\beta(s)$:

$$a_\beta(s) = \beta(s)a_{\text{reg}}(s) + (1 - \beta(s))a_{\text{rl}}(s), \ \ a_{\text{reg}}(s) = \pi_{\text{reg}}(s), a_{\text{rl}}(s) \sim \pi_{\text{rl}}(s). \tag{7}$$

**Lemma 1.** *(Policy Regularization) In any state $s \in \mathcal{S}$, for a multivariate Gaussian RL policy $\pi_{\text{rl}}$ with mean $\bar{\pi}_{\text{rl}}(s)$ and covariance matrix $\Sigma = \text{diag}(\sigma_1^2(s), \sigma_2^2(s), \ldots, \sigma_k^2(s)) \in \mathbb{R}^{k \times k}$, the expectation of the combined action $a_\beta(s)$ derived from Eq. (7) is the solution to the following regularized optimization with regularization parameter $\lambda = \beta(s)/(1 - \beta(s))$:*

$$\mathbb{E}\left[a_\beta(s)\right] = \underset{a}{\arg\min} \|a - \bar{\pi}_{\text{rl}}(s)\|_\Sigma + \frac{\beta(s)}{1 - \beta(s)} \|a - a_{\text{reg}}(s)\|_\Sigma. \tag{8}$$

We provide the proof of Lemma 1 in Appendix A.1, which is a state-dependent extension of the proof in [Cheng et al., 2019b]. Lemma 1 shows that the state-dependent $\beta(s)$ offers a safety mechanism on top of the safety regularizer. Since $\beta(s)$ is initialized close to 1 for $\forall s \in \mathcal{S}$, a strong regularization ($\lambda \to \infty$) from the safety regularizer policy is applied at the early stages of training. As learning progresses, the stochastic combined policy inevitably encounters rarely visited states, where $\pi_{\text{rl}}$ is poor due to the overestimated $Q$. However, the regularization parameter $\lambda$ remains large for these states, thus preventing the combined policy from safety violations by regularizing its deviation from the regularizer's safe policy. This deviation is quantified in the following theorem.

**Theorem 1.** *Assume the reward $R$ and the transition probability $P$ of the MDP $\mathcal{M}$ are Lipshitz continuous over $\mathcal{A}$ with Lipschitz constants $L_R$ and $L_P$. For any state $s \in \mathcal{S}$, the difference in expected*

*return between following the combined policy $\pi_\beta$ and following the safety regularizer policy $\pi_{\mathrm{reg}}$, i.e., $|V^{\pi_\beta}(s) - V^{\pi_{\mathrm{reg}}}(s)|$, has the upper-bound:*

$$|V^{\pi_\beta}(s) - V^{\pi_{\mathrm{reg}}}(s)| \leq \frac{(1-\gamma)|\mathcal{S}|L_R + \gamma|\mathcal{S}|L_P R_{max}}{(1-\gamma)^2}(1 - \beta(s))\Delta a, \qquad (9)$$

*where $|\mathcal{S}|$ is the cardinality of $\mathcal{S}$, $\Delta a = |a_{\mathrm{rl}}(s) - a_{\mathrm{reg}}(s)|$ is the bounded action difference at $s$.*

We provide the proof of Theorem 1 in Appendix A.2. Theorem 1 shows that when a state $s$ has not been sufficiently exploited and its corresponding $\beta(s)$ updates have been limited accordingly, the sub-optimality of the RL policy $\pi_{\mathrm{rl}}$ has limited impact on the expected return of the combined policy $\pi_\beta$, which is the actual acting policy. This is because $1 - \beta(s)$ remains close to zero at this stage, leading to only minor expected return deviations from the safety regularizer's policy $\pi_{\mathrm{reg}}$.

### 3.3 Updating the focus module

The focus module derives the policy combination (from the policies of safety regularizer and off-policy RL agent) that maximizes the expected return. For any state $s$, the state-dependent focus weight $\beta(s)$ is learned through updates according to the following objective:

$$\beta'(s) = \underset{\beta \in [0,1]}{\mathrm{argmax}}\, \mathbb{E}\left[Q^{\pi_\beta}(s, \beta a_{\mathrm{reg}}(s) + (1-\beta)a_{\mathrm{rl}}(s))\right], \; a_{\mathrm{reg}}(s) = \pi_{\mathrm{reg}}(s), a_{\mathrm{rl}}(s) \sim \pi_{\mathrm{rl}}(s). \quad (10)$$

Equation (10) is similar to the actor loss in actor-critic methods, however, instead of optimizing the policy network, Eq. (10) optimizes $\beta(s)$ for policy combination.

Compared to a scalar combination weight that applies the same policy combination across all states (e.g., as in [Cheng et al., 2019b]), the updated state-dependent weight $\beta'(s)$ in Eq. (10) guarantees monotonic performance improvement at least in the tabular cases, i.e., the update $\beta'(s_t)$ at a state $s_t$ results in $V^{\pi_{\beta'}}(s) \geq V^{\pi_\beta}(s)$ for all states $s \in \mathcal{S}$, where $\pi_{\beta'}$ is the combined policy proposed by $\beta'$. This can be proved by observing that the update in Eq. (10) results in a non-negative advantage for all states $s$, i.e., $Q^{\pi_\beta}(s, \pi_{\beta'}) \geq Q^{\pi_\beta}(s, \pi_\beta), \forall s \in \mathcal{S}$, where (with slight abuse of notation) we use $Q(s, \pi)$ to denote $Q(s, a)$ with $a \sim \pi(s)$. See Theorem 2 in Appendix A.3 for the detailed proof.

**Lemma 2.** *(Combination Weight Convergence) For any state $s$, assume the RL policy $\pi_{\mathrm{rl}}$ converges to the optimum policy $\pi^\star$ that satisfies $Q(s, \pi^\star) > Q(s, \pi), \forall \pi \neq \pi^\star$, then $\beta'(s) = 0$ will be the solution to Eq. (10) that achieves the optimal policy combination.*

Lemma 2 follows as $\pi_{\mathrm{reg}} \neq \pi^\star$ due to the sub-optimal model used to derive $\pi_{\mathrm{reg}}$. Let $a^\star(s) \sim \pi^\star(s)$ denote the optimum action at state $s$. If $\beta(s) \neq 0$, then $\beta(s)a_{\mathrm{reg}}(s) + (1 - \beta(s))a_{\mathrm{rl}}(s) = \beta(s)a_{\mathrm{reg}}(s) + (1 - \beta(s))a^\star(s) \neq a^\star(s)$. Therefore, the solution to Eq. (10), i.e., the updated focus weight $\beta'(s)$, can only be 0.

**Theorem 3.** *(Policy Combination Bias) For any state $s$, the distance between the combined action $a_\beta(s)$ and the optimal action $a^\star(s)$ has the following lower-bound:*

$$|a_\beta(s) - a^\star(s)| \geq |a_{\mathrm{reg}}(s) - a^\star(s)| - (1 - \beta(s))|a_{\mathrm{reg}}(s) - a_{\mathrm{rl}}(s)|. \qquad (11)$$

*If a Gaussian RL policy $\pi_{\mathrm{rl}}$ converges to the optimum policy $\pi^\star(s)$ with $Q(s, \pi^\star) > Q(s, \pi), \forall \pi \neq \pi^\star$, then the combined policy $\pi_\beta(s)$ can have unbiased convergence to the optimum Gaussian policy $\pi^\star$ with total variance distance $D_{\mathrm{TV}}(\pi_\beta(s), \pi^\star(s)) = 0$.*

The proof of Theorem 3 is given in Appendix A.5. Theorem 3 shows that by adaptively updating $\beta(s)$, the unbiased convergence of the combined policy can be achieved assuming i) a unique optimum solution and ii) the convergence of the RL agent, where the former follows naturally for most real-life control applications and the latter is well-established in the RL literature (the convergence of the specific RL agent used in RL-AR was proved in [Haarnoja et al., 2018]).

Algorithm 1 shows the pseudo-code of RL-AR, where $Q$, $\pi_{\mathrm{rl}}$, and $\beta$ are approximated with neural networks for practical applications with large or continuous state space. Note that the policy regularization (Lemma 1) and the convergence of RL-AR to the optimum RL policy (Lemma 2 and Theorem 3) still hold when using function approximation. For the RL agent, we take the standard approach of approximating $Q^\pi$ and $\pi$ with neural networks $Q_\phi$ and $\pi_\theta$. The focus module $\beta(s)$ is approximated with a neural network $\beta_\psi$ with outputs scaled to the range $(0, 1)$. Before learning

begins, $\beta_\psi$ is pretrained to output values close to 1 (e.g., $\beta_\psi(s) \geq 1 - \epsilon$) for all states to prioritize the safe $\pi_{\text{reg}}$. While interacting with the environment, each transition $e = (s, a, s', r, d, a_{\text{reg}})$ is stored in a replay buffer $\mathcal{D}$. Since $\pi_{\text{reg}}$ is deterministic and not subject to updates, by storing the action term $a_{\text{reg}}$ in $\mathcal{D}$, the optimization problem in Eq. (6) needs to be solved only once for any state $s$, significantly lowering the computational cost.

In Algorithm 1, details of $Q_{\phi_i}$ and $\pi_\theta$ updates (lines 8-9) are omitted as they follow the standard SAC [Haarnoja et al., 2018] paradigm elaborated in Section 2. After updating $Q_{\phi_i}$ and $\pi_\theta$, the focus module $\beta_\psi$ is updated using samples $(s, a_{\text{reg}})$ from replay buffer $\mathcal{D}$. As shown in line 10, $\beta_\psi$ is updated using gradient ascent with the gradient:

$$\nabla_\psi \frac{1}{|\mathcal{B}|} \sum_{(s, a_{\text{reg}}) \in \mathcal{B}} \min_{i=1,2} Q_{\phi_i}\left(s, \beta_\psi(s)a_{\text{reg}} + (1 - \beta_\psi(s))a_{\text{rl}}(s)\right), \quad a_{\text{rl}}(s) \sim \pi_\theta(s). \quad (12)$$

Note that the updated $Q_{\phi_i}, i = 1, 2$, and $\pi_\theta$ are used in Eq. (12) to allow quick response to new information. The clipped double-Q learning (taking the minimum $Q_{\phi_i}, i = 1, 2$) mitigates the overestimation error. Although exploitation level is not explicitly considered in Eq. (10), the use of replay buffer and the gradient-based updates in Eq. (12) mean more frequently-visited states with well-estimated Q values will affect $\beta_\psi(s)$ more, whereas rarely-visited states with overestimated Q values affect $\beta_\psi(s)$ less.

## 4  Numerical Experiments

Here, RL-AR is validated in critical settings described in Section 1, where the actual environment model $P$ is unknown, but an estimated environment model $\tilde{f}$ is available (e.g., from previous observations in the system or a similar system)[1]. Four safety-critical environments are implemented:

- *Glucose* is the critical medical control problem of regulating blood glucose level against meal-induced disturbances [Batmani, 2017]. The observations are denoted as $(G, \dot{G}, t)$, where $G$ is the blood glucose level, $\dot{G} = G_t - G_{t-1}$, and $t$ is the time passed after meal ingestion. The action is insulin injection, denoted as $a_I$. Crossing certain safe boundaries of $G$ can lead to catastrophic health consequences (hyperglycemia or hypoglycemia).

- *BiGlucose* is similar to the Glucose environment but capturing more complicated blood glucose dynamics, with 12 internal states (11 unobservable), 2 actions with large delays, and nondifferentiable piecewise dynamics. [Kalisvaart et al., 2023]. The observations are the same as Glucose. The actions are insulin and glucagon injections, denoted as $(a_I, a_N)$.

- *CSTR* is a continuous stirred tank reactor for regulating the concentration of a chemical $C_B$ [Fiedler et al., 2023]. The observations are $(C_A, C_B, T_R, T_K)$, where $C_A$ and $C_B$ are the concentrations of two chemicals; $T_R$ and $T_K$ are the temperatures of the reactor and the cooler, respectively. The actions are the feed and the heat flow, denoted as $(a_F, a_Q)$. Crossing safe boundaries of $C_A$, $C_B$, and $T_R$ can lead to tank failure or even explosions.

- *Cart Pole* is a classic control problem of balancing an inverted pole on a cart by applying horizontal force to the cart. The environment is adapted from the gymnasium environment [Towers et al., 2023] with continuous action space. The observations are $(x, \dot{x}, \theta, \dot{\theta})$, where $x$ is the position of the cart, $\theta$ is the angle of the pole, $\dot{x} = x_t - x_{t-1}$, and $\dot{\theta} = \theta_t - \theta_{t-1}$. The action is the horizontal force, denoted as $a_f$. The control fails if the cart reaches the end of its rail or the pole falls over.

All environments are simulated following widely accepted models and parameters [Sherr et al., 2022, Yang and Zhou, 2023], which are assumed to be unknown to the control algorithm. The estimated models and the actual environments are set to have different model parameters. For *Glucose* and *BiGlucose*, the estimated model parameters are derived from real patient measurements [Hovorka et al., 2004, Zahedifar and Keymasi Khalaji, 2022]. The environment models, parameters, and reward functions are detailed in Appendix B.

The baseline methods used in the experiments are: i) MPC [Fiedler et al., 2023], the primary method for control applications with safety constraints [Hewing et al., 2020]; ii) SAC [Haarnoja et al., 2018],

---

[1]Code available at `https://github.com/HaozheTian/RL-AR`.

a model-free RL that disregards safety during training, but achieves state-of-the-art normalized returns; iii) Residual Policy Learning (RPL) [Silver et al., 2018], an RL method that improves a sub-optimal MPC policy by directly applying a residual policy action; iv) Constrained Policy Optimization (CPO) [Achiam et al., 2017], a widely-used risk-aware safe RL benchmark based on the trust region method; and v) SEditor [Yu et al., 2022b], a more recent, state-of-the-art safe RL method that learns a safety editor for transforming potentially unsafe actions.

The proposed method, RL-AR, uses MPC as the safety regularizer agent and SAC as the off-policy RL agent. The two agents in RL-AR each follow their respective baseline implementations. The focus module in RL-AR has a $[128, 32]$ hidden layer size with ReLU activation, and $k$ outputs scaled to $(0, 1)$ by a shifted $\tanh$. Additional detail and hyperparameters of the implementations are provided in Appendix C. Our RL-AR implementation has an average decision and update time of 0.037 seconds per step on a laptop with a single GPU, meeting real-time control requirements across all environments. In Appendix D we present ablation studies on the benefit of state-dependent focus weight and the choice of SAC as the RL agent.

## 4.1 Safety of training

We begin by evaluating training safety in the actual environment by counting the number of failed episodes out of the first 100 training episodes. An episode is considered a failure and terminated immediately if a visited state exceeds a predefined safety bound. As shown in Table 1, only RL-AR completely avoids failure during training in the actual environment. Although MPC does not fail, it does not adapt or update its policy in the actual environ-

Table 1: The mean ($\pm$ standard deviation) number of failures out of the first 100 training episodes, obtained over 5 runs with different random seeds.

| Method | Gluc. | BiGl. | CSTR | Cart. |
|---|---|---|---|---|
| RL-AR | **0.0** (0.0) | **0.0** (0.0) | **0.0** (0.0) | **0.0** (0.0) |
| MPC | **0.0** (0.0) | **0.0** (0.0) | **0.0** (0.0) | **0.0** (0.0) |
| SAC | 19.0 (15.2) | 59.4 (31.1) | 99.2 (0.4) | 93.6 (7.3) |
| RPL | 7.8 (6.4) | 5.6 (3.9) | 3.6 (1.5) | 3.6 (2.2) |
| CPO | 8.0 (2.1) | 72.4 (6.7) | 100.0 (0.0) | 21.8 (3.7) |
| SEditor | 6.8 (1.7) | 74.6 (8.4) | 97.2 (5.6) | 17.4 (10.6) |

ment. RPL is relatively safe by relying on a safe initial policy, but its un-regularized residual policy action results in less stable combined action, leading to failures. Due to their model-free nature, CPO and SEditor must observe failures in the actual environment before learning a safe policy, thus failing many times during training. Note that SAC averages the largest number of failures over all environments.

Next, since the estimated environment model $\tilde{f}$ is integrated into RL-AR, MPC, and RPL, for a fair comparison we pretrain the model-free SAC, CPO, and SEditor using $\tilde{f}$ as an environment simulator; this allows all methods to access the estimated model before training on the actual environment. Figure 2 compares the normalized episodic return and the number of failures for different methods over training episodes; the proposed method is compared with SAC and RPL in Fig. 2A, and with MPC, CPO, and SEditor (safety-aware methods) in Fig. 2B. The mean (solid lines) and standard deviation (shaded area) in Fig. 2 are obtained from 5 independent runs using different random seeds. Episodes are terminated on failure, resulting in varying episode lengths, thus, the episodic returns are normalized by episode lengths.

Two important insights can be drawn from Fig. 2. First, the normalized return curves show that RL-AR consistently achieves higher returns faster than other methods across all environments. RL-AR begins with a reliable initial policy derived from the safety regularizer and incrementally integrates a learned policy, resulting in stable return improvements (as suggested by Lemma 1 and Theorem 1). RL-AR shows a steady return improvement, except for some fluctuations in the CSTR environment which the method quickly recovers from. In contrast, the baseline methods—SAC and RPL, which apply drastic actions based on overestimated returns, or CPO and SEditor, which impose constraints using biased cost estimates derived from simulations using $\tilde{f}$—exhibit significant return degradation and even failures. Second, RL-AR effectively avoids failure during training (see the bottom rows in Fig. 2A&B). Note that pertaining on $\tilde{f}$ leads to fewer failures in the actual environment for SAC, CPO, and SEditor (compare with the results in Table 1). However, SAC, CPO, and SEditor continue to fail despite the pretraining (the only exception is SEditor in the Glucose environment), indicating that pretraining on estimated model is not an effective approach to achieve safety.

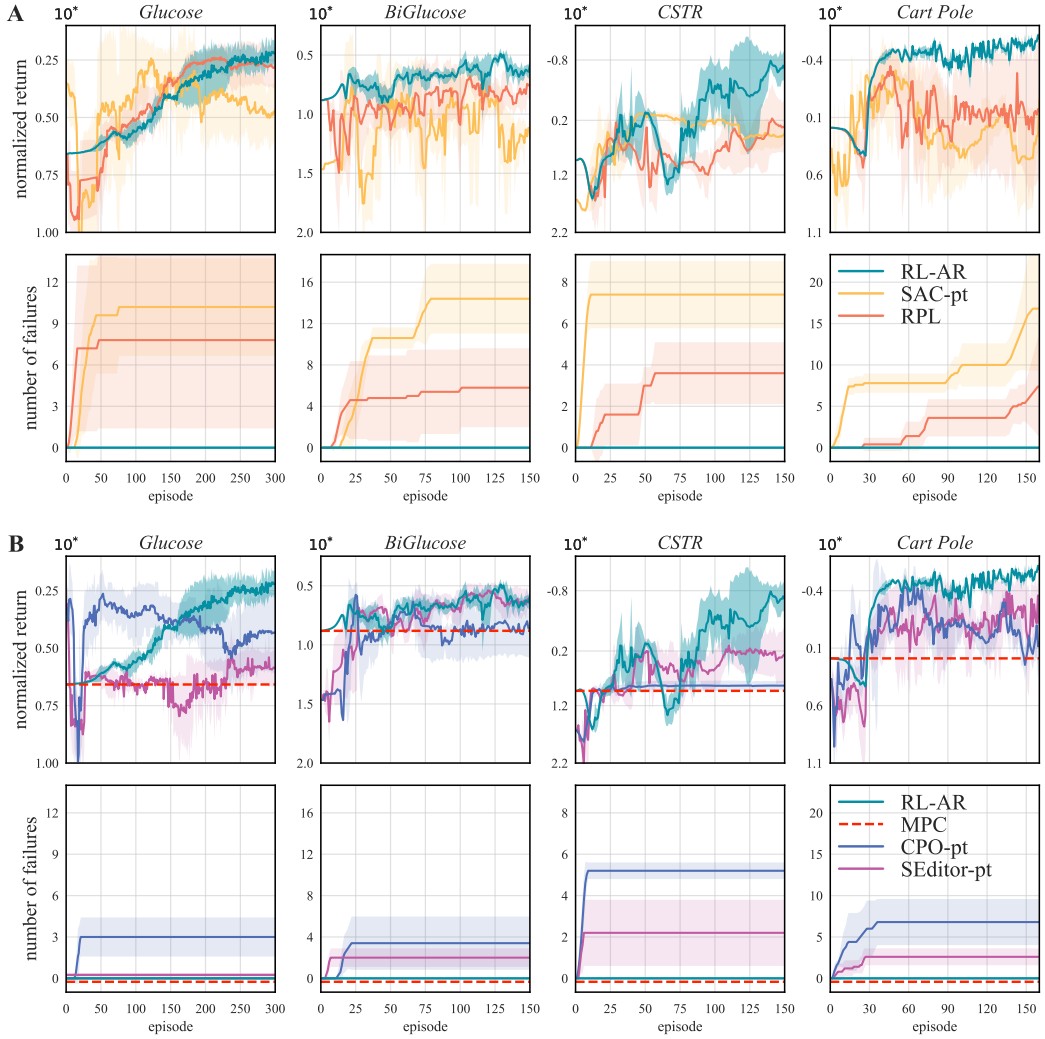

Figure 2: The normalized return curves and the number of failures during training (standard deviations are shown in the shaded areas). SAC, CPO, and SEditor are pretrained using the estimated model $\tilde{f}$ as a simulator (as indicated by "-pt") to ensure a fair comparison, given that RL-AR, MPC, and RPL inherently incorporate the estimated model. This pretraining allows SAC, CPO, and SEditor to leverage the estimated model, resulting in more competitive performance in the comparison.

In CSTR and Cart Pole environments, and only in a limited number of episodes during the early stages of training, the proposed RL-AR policy's normalized return falls below that of the static MPC policy. This can occur due to RL-AR reaching insufficiently learned states (with overestimated Q values). Nevertheless, since $\beta(s)$ is close to 1 for these insufficiently learned states, the dominance of the safety regularizer agent allows RL-AR to converge to high returns without compromising the safety (as shown in Theorem 1).

## 4.2 Achieved return after convergence

Besides ensuring safer training, RL-AR theoretically enables unbiased convergence to the optimal RL policy (as shown in Theorem 2). We validate this by testing whether RL-AR matches the return of SAC. SAC is shown to consistently converge to well-performing control policies, competitive if not better than other state-of-the-art RL algorithms [Raffin et al., 2021, Huang et al., 2022]. In Fig. 3, we compare the control trajectories of RL-AR, SAC, and MPC and the returns of their converged policies after training; we run the converged policies without stochastic exploration. RL-AR significantly outperforms MPC, achieving faster regulation, reduced oscillation, and smaller steady-

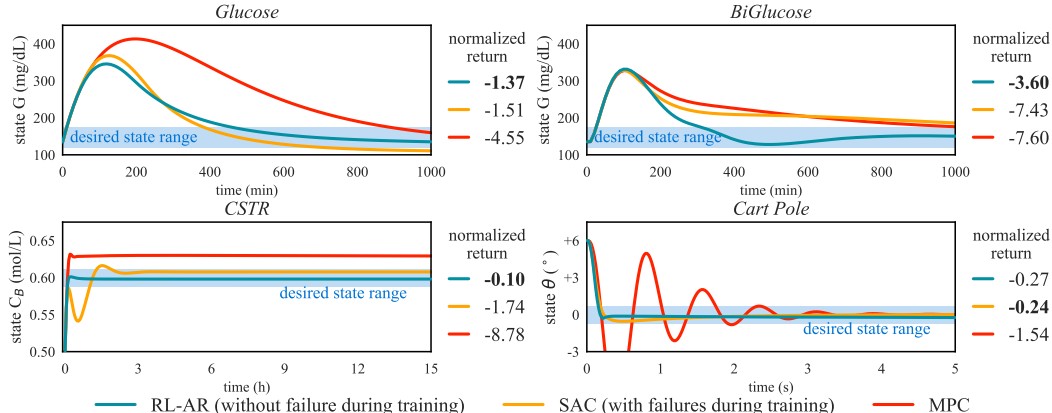

Figure 3: Comparison of the converged trajectories and their corresponding normalized return. In the upper row, the agents try to retain the desired state under time-varying disturbances; in the lower row, the agents try to steer the system to a desired state. Although SAC fails before converging, here we compare with the converged SAC results to show that RL-AR can achieve the performance standard of model-free RL that prioritizes return and disregards safety.

state error. Furthermore, in terms of normalized return, RL-AR is competitive with SAC in the Cart Pole environment and outperforms SAC in the other three environments. The results demonstrate that RL-AR not only effectively ensures safety during training, but also finds control policies competitive with the state-of-the-art model-free RL.

### 4.3 Sensitivity to parameter discrepancies

Inherently, RL-AR's training safety relies on the effectiveness of the safety regularizer, which depends on the quality of the estimated model $\tilde{f}$. Thus, a large discrepancy between $\tilde{f}$ and the actual environment might compromise the training safety of RL-AR. We empirically quantify this effect by deploying RL-AR in discrepant Glucose environments created by varying the environment model parameters $n$ and $p_2$ (values chosen based on $\tilde{n}$ and $\tilde{p}_2$ in $\tilde{f}$) to mimic deviating characteristics of new patients, and counting the number of failed episodes out of the first 100 episodes in Fig. 4. Lower $p_2/\tilde{p}_2$ and $n/\tilde{n}$ makes the environment more susceptible to failure; see Appendix B.1. The results show that RL-AR can withstand reasonable discrepancies between $\tilde{f}$ and the actual environment. Failures only occur when the actual environment deviates significantly from $\tilde{f}$ with $p_2 \leq \frac{3}{8}\tilde{p}_2$ and $n \leq \frac{6}{16}\tilde{n}$. All failures are caused by the safety regularizer due to its misleading estimated model with largely discrepant parameters. When RL adapts (by updating $\pi_\theta$ and $\beta_\psi$) sufficiently to correct the misleading regularizer action, the combined agents effectively recover from failure. Here in our tests in the Glucose environment, even with large model discrepancies, RL-AR is shown to be as safe as the classic MPC. Appendix Fig. 7 provides insights into the adaptation of the focus module by showing the progression of $\beta_\psi$ in the learning process.

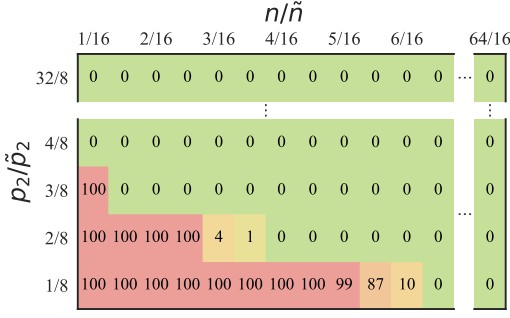

Figure 4: Number of failed training episodes out of the first 100 in Glucose environment with different degrees of parameter discrepancy.

## 5 Related Works

The existing safe RL works can be roughly divided into two categories [García and Fernández, 2015]. The first category does not require knowledge of the system dynamics. These methods often rely on probabilistic approaches [Geibel and Wysotzki, 2005, Liu et al., 2022] or constrained

MDP [Achiam et al., 2017, Yang et al., 2020]. More recent methods use learned models to filter out unsafe actions [Bharadhwaj et al., 2020]. However, these methods need to observe failures to estimate the safety cost, thus do not ensure safety during training. This category of methods does not apply to the single-life setting in this work, i.e., no failure is tolerated in the actual environment. Nevertheless, in Fig. 2 we evaluate pertaining CPO [Achiam et al., 2017] and SEditor [Yu et al., 2022b] using simulation with the estimated model to obtain risk estimation before training in the actual environment.

The second category relies on an estimated model of the system dynamics. Some methods enforce safety constraints using Control Barrier Function (CBF) [Cheng et al., 2019a]. However, CBF minimizes the control effort without directly optimizing the system performance. In contrast, the MPC regularizer used in RL-AR enforces safety constraints while optimizing the predicted performance, resulting in high performance during training. Some methods compute a model-based projection to verify the safety of actions [Bastani, 2021, Kochdumper et al., 2023, Fulton and Platzer, 2018]. However, the scalability of verification-based methods for complex control applications is an issue. Anderson et al. [2020] propose using neurosymbolic representations to reduce verification complexity, but the computational cost remains to be high. On the other hand, the average time for RL-AR to take a step (including the environment interaction and network updates) in the four environments in Section 4 is 0.037 s, which is practical for real-time control.

Gros and Zanon [2019] and Zanon et al. [2020] use MPC as the policy generator and use RL to dynamically tune the MPC parameters in the cost functions and the estimated environment model. Assuming discrepancies between the estimated model parameters and the actual environment parameters, the tuning increases the MPC's performance. However, this is a strong assumption since there are other discrepancies, such as neglected dynamics and discretization errors. However, the RL-AR proposed in this work can theoretically converge to the optimal policy by utilizing the model-free RL agent.

It is important to note that although the MPC regularizer accelerates the learning of the RL agent, RL-AR is not a special case of transferring a learned policy. The MPC regularizer used in our proposed algorithm forecasts the system behavior and hard-codes safety constraints in the optimization. The main role of the MPC is to keep the RL-AR actions safe in the actual environment—not transferring knowledge. Transfer learning in RL studies the effective reuse of knowledge, especially across different tasks [Taylor and Stone, 2009, Glatt et al., 2020]. By reusing prior knowledge, transferred RL agents skip the initial random trial-and-error and drastically increase sampling efficiency [Karimpanal et al., 2020, Da Silva and Costa, 2019]. However, transferred RL agents are not inherently risk-aware, and thus can still steer the actual environment into unsafe states. For this reason, transferred RL is not generally considered effective for ensuring safety.

# 6 Conclusion and Future Works

Controlling critical systems, where unsafe control actions can have catastrophic consequences, has significant applications in various disciplines from engineering to medicine. Here, we demonstrate that the appropriate combination of a control regularizer can facilitate safe RL. The proposed method, RL-AR, learns a focus module that relies on the safe control regularizer for less-exploited states and simultaneously allows unbiased convergence for well-exploited states. Numerical experiments in critical applications revealed that RL-AR is safe during training, given a control regularizer with reasonable safety performance. Furthermore, RL-AR effectively learns from interactions and converges to the performance standard of model-free RL that disregards safety.

One limitation of our setting is the assumption that the estimated model has reasonable accuracy for deriving a viable control regularizer. Although this assumption is common in the control and safe RL literature, one possible direction for future work is to design more robust algorithms against inaccurate estimated models of the actual environment. A potential approach is to update the estimated model using observed transitions in the actual environment. However, the practical challenge is to adequately adjust all model parameters even with a small number of transitions observed in the actual environment. In addition, for such an approach, managing controllability, convergence, and safety requires careful design and tuning.

## Acknowledgments and Disclosure of Funding

This project is partly supported by UK Research and Innovation (UKRI) under the UK government's Horizon Europe funding guarantee [grant number 101084642].

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

# Appendix

## Table of Content

## A Theoretical analysis

### A.1 Policy combination as regularization

**Lemma 1.** *(Policy Regularization) In any state $s \in \mathcal{S}$, for a multivariate Gaussian RL policy $\pi_{\mathrm{rl}}$ with mean $\bar{\pi}_{\mathrm{rl}}(s)$ and covariance matrix $\Sigma = \mathrm{diag}(\sigma_1^2(s), \sigma_2^2(s), \ldots, \sigma_k^2(s)) \in \mathbb{R}^{k \times k}$, the expectation of the combined action $a_\beta(s)$ derived from Eq. (7) is the solution to the following regularized optimization with regularization parameter $\lambda = \beta(s)/(1 - \beta(s))$:*

$$\mathbb{E}\left[a_\beta(s)\right] = \underset{a}{\arg\min} \|a - \bar{\pi}_{\mathrm{rl}}(s)\|_\Sigma + \frac{\beta(s)}{1 - \beta(s)} \|a - a_{\mathrm{reg}}(s)\|_\Sigma. \tag{13}$$

*Proof.* For state $s \in \mathcal{S}$, the focus module outputs a fixed $\beta = \beta(s)$. For the fixed $\beta$, the proof of Lemma 1 is similar to the proof by Cheng et al. [2019b]. Since the RL policy is a Gaussian distributed policy $\mathcal{N}(\bar{\pi}_{\mathrm{rl}}(s), \Sigma)$ with the mean action $\bar{\pi}_{\mathrm{rl}}(s)$ and an exploration noise with covariance $\Sigma = \mathrm{diag}(\sigma_1^2, \sigma_2^2, \ldots)$, the combined action $a_\beta(s)$ also follows a Gaussian distribution:

$$a_\beta(s) \sim \mathcal{N}\left(\beta a_{\mathrm{reg}}(s) + (1 - \beta)\bar{\pi}_{\mathrm{rl}}(s), (1 - \beta)^2 \Sigma\right). \tag{14}$$

Let $f(\mu, \Sigma)$ be the probability density function (PDF) of $\mathcal{N}(\mu, \Sigma)$. The product of two multivariate Gaussian PDFs is proportional to another multivariate Gaussian PDF with the following mean and covariance:

$$f\left(\mu_1, \Sigma_1\right) \cdot f\left(\mu_2, \Sigma_2\right) = cf\left((\Sigma_1^{-1} + \Sigma_2^{-1})^{-1}(\Sigma_1^{-1}\mu_1 + \Sigma_2^{-1}\mu_2), (\Sigma_1^{-1} + \Sigma_2^{-1})^{-1}\right). \tag{15}$$

The mean of $a_\beta(s)$, $\beta a_{\mathrm{reg}}(s) + (1 - \beta)\bar{\pi}_{\mathrm{rl}}(s)$, can be expressed in the following form:

$$\beta a_{\mathrm{reg}}(s) + (1 - \beta)\bar{\pi}_{\mathrm{rl}}(s) = \beta \Sigma^{-1} \Sigma a_{\mathrm{reg}}(s) + (1 - \beta)\Sigma^{-1}\Sigma \bar{\pi}_{\mathrm{rl}}(s)$$
$$= \Sigma\left(\left(\frac{1}{\beta}\Sigma\right)^{-1} a_{\mathrm{reg}}(s) + \left(\frac{1}{1-\beta}\Sigma\right)^{-1} \bar{\pi}_{\mathrm{rl}}(s)\right). \tag{16}$$

The covariance matrix $\Sigma$ can be expanded into the following form:

$$\Sigma = \left(\left(\frac{1}{\beta}\Sigma\right)^{-1} + \left(\frac{1}{1-\beta}\Sigma\right)^{-1}\right)^{-1}. \tag{17}$$

Using Eq. (15), the PDF of $a_\beta(s)$ can be expressed as the multiplication of two PDFs, as shown below:

$$f(\beta a_{\mathrm{reg}}(s) + (1 - \beta)\bar{\pi}_{\mathrm{rl}}(s), (1 - \beta)^2 \Sigma) = c_0 f\left(a_{\mathrm{reg}}(s), \frac{1}{\beta}\Sigma\right) \cdot f\left(\bar{\pi}_{\mathrm{rl}}(s), \frac{1}{1-\beta}\Sigma\right). \tag{18}$$

With the definition $\|x\|_\Sigma = x^T \Sigma^{-1} x$, the PDF of $a_\beta(s)$ can be written as:

$$f(a_\beta(s)) = c_0 c_1 \exp\left(-\frac{\beta}{2}\|a - a_{\mathrm{reg}}(s)\|_\Sigma\right) \times c_2 \exp\left(-\frac{1-\beta}{2}\|a - \bar{\pi}_{\mathrm{rl}}(s)\|_\Sigma\right)$$
$$= c \exp\left(\frac{1-\beta}{2}\left(-\|a - \bar{\pi}_{\mathrm{rl}}(s)\|_\Sigma - \frac{\beta}{1-\beta}\|a - a_{\mathrm{reg}}(s)\|_\Sigma\right)\right), \tag{19}$$

where the constant $c$ is as follows:

$$c = \frac{c_0}{(2\pi)^k \beta^{k/2}(1-\beta)^{k/2}|\Sigma|}. \tag{20}$$

Since $\beta$ is in the range $(0,1)$, $f(a_\beta(s))$ monotonically decreases as $\|a - \bar{\pi}_{\mathrm{rl}}(s)\|_\Sigma + \frac{\beta}{1-\beta}\|a - a_{\mathrm{reg}}(s)\|_\Sigma$ increases. Therefore, the probability of $\pi(s)$ is maximized when the term is minimized, leading to the following optimization problem:

$$\mathbb{E}\left[a_\beta(s)\right] = \underset{a}{\operatorname{argmin}} \|a - \bar{\pi}_{\mathrm{rl}}(s)\|_\Sigma + \frac{\beta(s)}{1-\beta(s)}\|a - a_{\mathrm{reg}}(s)\|_\Sigma. \tag{21}$$

Assuming the RL policy $\pi_{\mathrm{rl}}$ converges to $\operatorname{argmax}_\pi Q(s, \pi(s))$, Eq. (21) can be written as:

$$\mathbb{E}\left[a_\beta(s)\right] = \underset{a}{\operatorname{argmin}} \left\|a - \mathbb{E}\left[\underset{\pi}{\operatorname{argmax}} Q(s, \pi(s))\right]\right\|_\Sigma + \frac{\beta(s)}{1-\beta(s)}\|a - a_{\mathrm{reg}}(s)\|_\Sigma. \tag{22}$$

$\square$

## A.2 Deviation of combined policy from safety regularizer

**Theorem 1.** *Assume the reward $R$ and the transition probability $P$ of the MDP $\mathcal{M}$ are Lipshitz continuous over $\mathcal{A}$ with Lipschitz constants $L_R$ and $L_P$. For any state $s \in \mathcal{S}$, the difference in expected return between following the combined policy $\pi_\beta$ and following the safety regularizer policy $\pi_{\mathrm{reg}}$, i.e., $|V^{\pi_\beta}(s) - V^{\pi_{\mathrm{reg}}}(s)|$, has the upper-bound:*

$$|V^{\pi_\beta}(s) - V^{\pi_{\mathrm{reg}}}(s)| \leq \frac{(1-\gamma)|\mathcal{S}|L_R + \gamma|\mathcal{S}|L_P R_{max}}{(1-\gamma)^2}(1-\beta(s))\Delta a, \tag{23}$$

*where $|\mathcal{S}|$ is the cardinality of $\mathcal{S}$, and $\Delta a = |a_{\mathrm{rl}}(s) - a_{\mathrm{reg}}(s)|$ is the bounded action difference at $s$.*

*Proof.* Let us define value function vectors $\mathbf{v}_\beta$ and $\mathbf{v}_{\mathrm{reg}}$ in $\mathbb{R}^{|\mathcal{S}|\times 1}$, for which the $s$-th entries are $[\mathbf{v}_\beta]_s = V^{\pi_\beta}(s)$ and $[\mathbf{v}_{\mathrm{reg}}]_s = V^{\pi_{\mathrm{reg}}}(s)$. Also, let us by $\mathbf{r}_\beta$ and $\mathbf{r}_{\mathrm{reg}}$ denote reward vectors in $\mathbb{R}^{|\mathcal{S}|\times 1}$, with the $s$-th entries of the reward vectors being $[\mathbf{r}_\beta]_s = r(s, a_\beta(s))$ and $[\mathbf{r}_{\mathrm{reg}}]_s = r(s, a_{\mathrm{reg}}(s))$. We define state-transition matrices $\mathbf{P}_\beta$ and $\mathbf{P}_{\mathrm{reg}}$ in $\mathbb{R}^{|\mathcal{S}|\times|\mathcal{S}|}$, with $(s, s')$-th entries as $[\mathbf{P}_\beta]_{s,s'} = P(s, a_\beta(s))$ and $[\mathbf{P}_{\mathrm{reg}}]_{s,s'} = P(s, a_{\mathrm{reg}}(s))$. According to the vectorized Bellman equation, the difference between $\mathbf{v}_\beta$ and $\mathbf{v}_{\mathrm{reg}}$ satisfies the following relationship:

$$\begin{aligned}
\mathbf{v}_\beta - \mathbf{v}_{\mathrm{reg}} &= \mathbf{r}_\beta + \gamma\mathbf{P}_\beta\mathbf{v}_\beta - \mathbf{r}_{\mathrm{reg}} - \gamma\mathbf{P}_{\mathrm{reg}}\mathbf{v}_{\mathrm{reg}} \\
&= \mathbf{r}_\beta - \mathbf{r}_{\mathrm{reg}} + \gamma\mathbf{P}_\beta\mathbf{v}_\beta - \gamma\mathbf{P}_\beta\mathbf{v}_{\mathrm{reg}} + \gamma\mathbf{P}_\beta\mathbf{v}_{\mathrm{reg}} - \gamma\mathbf{P}_{\mathrm{reg}}\mathbf{v}_{\mathrm{reg}} \\
&= \mathbf{r}_\beta - \mathbf{r}_{\mathrm{reg}} + \gamma\mathbf{P}_\beta(\mathbf{v}_\beta - \mathbf{v}_{\mathrm{reg}}) + \gamma(\mathbf{P}_\beta - \mathbf{P}_{\mathrm{reg}})\mathbf{v}_{\mathrm{reg}} \\
&= (\mathbf{I} - \gamma\mathbf{P}_\beta)^{-1}(\mathbf{r}_\beta - \mathbf{r}_{\mathrm{reg}} + \gamma(\mathbf{P}_\beta - \mathbf{P}_{\mathrm{reg}})\mathbf{v}_{\mathrm{reg}})
\end{aligned} \tag{24}$$

Let $\mathbf{d}_{\beta,s}^T$ be the $s$-th row of $(\mathbf{I} - \gamma\mathbf{P}_\beta)^{-1}$ and $\mathbf{d}_{\mathrm{reg},s}^T$ be the $s$-th row of $(\mathbf{I} - \gamma\mathbf{P}_{\mathrm{reg}})^{-1}$. Since $(\mathbf{I} - \gamma\mathbf{P}_\beta)^{-1}$ can be expanded as a Neumann series $I + \gamma P_\beta + \gamma^2 P_\beta^2 \cdots$, the upper-bound for the elements of $\mathbf{d}_{\beta,s}$ and $\mathbf{d}_{\mathrm{reg},s}$ is $1/(1-\gamma)$, i.e., $\|\mathbf{d}_{\beta,s}\|_\infty$ and $\|\mathbf{d}_{\mathrm{reg},s}\|_\infty$ are less than or equal to $1/(1-\gamma)$. From Eq. (24), the value functions $V^{\pi_\beta}(s)$ and $V^{\pi_{\mathrm{reg}}}(s)$ for a specific state $s \in \mathcal{S}$ satisfies:

$$|V^{\pi_\beta}(s) - V^{\pi_{\mathrm{reg}}}(s)| \leq \underbrace{|\mathbf{d}_{\beta,s}^T(\mathbf{r}_\beta - \mathbf{r}_{\mathrm{reg}})|}_{(a)} + \underbrace{\gamma|\mathbf{d}_{\mathrm{reg},s}^T(\mathbf{P}_\beta - \mathbf{P}_{\mathrm{reg}})\mathbf{v}_{\mathrm{reg}}|}_{(b)}, \tag{25}$$

First, we consider part $(a)$ of Eq. (25). Assuming $R(s, a)$ is Lipschitz continuous over $\mathcal{A}$ with Lipschitz constant $L_R$, we have:

$$\begin{aligned}
\|\mathbf{r}_\beta - \mathbf{r}_{\mathrm{reg}}\|_1 &= |\mathcal{S}||R(s, a_\beta(s)) - R(s, a_{\mathrm{reg}}(s_t))| \\
\text{(Lipschitz)} \quad &\leq |\mathcal{S}|L_R(1-\beta(s))|a_{\mathrm{rl}}(s) - a_{\mathrm{reg}}(s)| \\
&\leq |\mathcal{S}|L_R(1-\beta(s))\Delta a
\end{aligned} \tag{26}$$

Using the Holder's inequality, part (*a*) of Eq. (25) has the following upper-bound:

$$|\mathbf{d}_{\beta,s}^T(\mathbf{r}_\beta - \mathbf{r}_{\text{reg}})| \leq \|\mathbf{d}_{\beta,s}\|_\infty \|\mathbf{r}_\beta - \mathbf{r}_{\text{reg}}\|_1 \leq \frac{|\mathcal{S}|L_R}{1-\gamma}(1-\beta(s))\Delta a \tag{27}$$

Second, we consider part (*b*) of Eq. (25). For each state $s \in \mathcal{S}$, define the *s*-th rows of $\mathbf{P}_\beta$ and $\mathbf{P}_{\text{reg}}$ as $\mathbf{p}_{\beta,s}^T$ and $\mathbf{p}_{\text{reg},s}^T$. The vectors $\mathbf{p}_{\beta,s}$ and $\mathbf{p}_{\text{reg},s}$ each represents a discrete probability distribution. Because we assume $P(s,a)$ is Lipschitz continuous over $\mathcal{A}$ with Lipschitz constant $L_P$, the upper-bound on each item in vector $(\mathbf{P}_\beta - \mathbf{P}_{\text{reg}})\mathbf{v}_{\text{reg}}$ is derived below:

$$
\begin{aligned}
|(\mathbf{p}_{\beta,s} - \mathbf{p}_{\text{reg},s})^T \mathbf{v}_{\text{reg}}| &\leq \|\mathbf{p}_{\beta,s} - \mathbf{p}_{\text{reg},s}\|_1 \|\mathbf{v}_{\text{reg}}\|_\infty \\
&\leq \frac{R_{\max}}{1-\gamma}\|P(s,a_\beta(s)) - P(s,a_{\text{reg}}(s))\|_1 \\
\text{\color{blue}(Lipschitz)} \quad &\leq \frac{L_P R_{\max}}{1-\gamma}(1-\beta(s))|a_{\text{rl}}(s) - a_{\text{reg}}(s)| \\
&\leq \frac{L_P R_{\max}}{1-\gamma}(1-\beta(s))\Delta a
\end{aligned}
\tag{28}
$$

Therefore, part (*b*) of Eq. (25) has the following upper-bound:

$$\gamma|\mathbf{d}_{\text{reg},s}^T(\mathbf{P}_\beta - \mathbf{P}_{\text{reg}})\mathbf{v}_{\text{reg}}| \leq \|\mathbf{d}_{\text{reg},s}\|_\infty \|(\mathbf{P}_\beta - \mathbf{P}_{\text{reg}})\mathbf{v}_{\text{reg}}\|_1 \leq \frac{\gamma|\mathcal{S}|L_P R_{\max}}{(1-\gamma)^2}(1-\beta(s))\Delta a \tag{29}$$

The proof is complete by substituting Eq. (27) and Eq. (29), respectively, into parts (*a*) and (*b*) of Eq. (25). □

## A.3 Performance improvement of focus module update

**Theorem 2.** *(Focus Module Performance Improvement) The focus weight $\beta'(s)$ updated by Eq. (10) satisfies $V^{\pi_{\beta'}}(s) \geq V^{\pi_\beta}(s), \forall s \in \mathcal{S}$, i.e., the expected return monotonically improves.*

*Proof.* According to the performance difference lemma in [Kakade and Langford, 2002], in all states $s \in \mathcal{S}$, the expected return difference between the two policies satisfies:

$$V^{\pi'}(s) - V^\pi(s) = \frac{1}{1-\gamma}\mathbb{E}_{s' \sim d^{\pi',s}}[A^\pi(s', \pi')], \quad \forall \pi, \pi' \tag{30}$$

where $A^\pi(s, \pi') = Q^\pi(s, \pi') - Q^\pi(s, \pi)$ is the advantage function, $d^{\pi',s}$ is the normalized discounted occupancy induced by policy $\pi'$ from the starting state $s$. The update in Eq. (10) results in $Q^{\pi_\beta}(s, \pi_{\beta'}) \geq Q^{\pi_\beta}(s, \pi_\beta), \forall s \in \mathcal{S}$. Thus $\beta'(s)$ satisfies:

$$A^{\pi_\beta}(s, \pi_{\beta'}) = Q^{\pi_\beta}(s, \pi_{\beta'}) - Q^{\pi_\beta}(s, \pi_\beta) \geq 0. \tag{31}$$

Using the above in Eq. (30), and combined with the fact that the discount factor $\gamma$ is in range $(0, 1)$, we can rewrite Eq. (30) as $V^{\pi_{\beta'}}(s) \geq V^{\pi_\beta}(s), \forall s \in \mathcal{S}$. □

## A.4 Convergence of combination weight

**Lemma 2.** *(Combination Weight Convergence) For any state s, assume the RL policy $\pi_{\text{rl}}$ converges to the optimum policy $\pi^\star$ that satisfies $Q(s, \pi^\star) > Q(s, \pi), \forall \pi \neq \pi^\star$, then $\beta'(s) = 0$ will be the solution to Eq. (10) that achieves the optimal policy combination.*

*Proof.* Since a sub-optimal model is used to derive $\pi_{\text{reg}}$, $\pi_{\text{reg}} \neq \pi^\star$. Let $a^\star(s) \sim \pi^\star(s)$ denote the optimum action at state $s$. If $\beta(s) \neq 0$, then $\beta(s)a_{\text{reg}}(s) + (1 - \beta(s))a_{\text{rl}}(s) = \beta(s)a_{\text{reg}}(s) + (1 - \beta(s))a^\star(s) \neq a^\star(s)$. Therefore, the solution to Eq. (10), i.e., the updated focus weight $\beta'(s)$, can only be 0. □

## A.5 Convergence to the RL policy

**Theorem 3.** *(Policy Combination Bias) For any state $s$, the distance between the combined action $a_\beta(s)$ and the optimal action $a^\star(s)$ has the following lower-bound:*

$$|a_\beta(s) - a^\star(s)| \geq |a_{\text{reg}}(s) - a^\star(s)| - (1 - \beta(s))|a_{\text{reg}}(s) - a_{\text{rl}}(s)|. \tag{32}$$

*If a Gaussian RL policy $\pi_{\text{rl}}$ converges to the optimum policy $\pi^\star(s)$ with $Q(s, \pi^\star) > Q(s, \pi), \forall \pi \neq \pi^\star$, then the combined policy $\pi_\beta(s)$ can have unbiased convergence to the optimum Gaussian policy $\pi^\star$ with total variance distance $D_{\text{TV}}(\pi_\beta(s), \pi^\star(s)) = 0$.*

*Proof.* First, we prove the lower-bound for $|a_\beta(s) - a^\star(s)|$. The proof of lower-bound is similar to [Cheng et al., 2019b]. We expand the distance between the safety regularizer action $a_{\text{reg}}$ and the combined action $a_\beta$ as follows:

$$\begin{aligned}
|a_{\text{reg}}(s) - a_\beta(s)| &= |a_{\text{reg}}(s) - \beta(s)a_{\text{reg}}(s) - (1 - \beta(s))a_{\text{rl}}(s)| \\
&= (1 - \beta(s))|a_{\text{reg}}(s) - a_{\text{rl}}(s)|.
\end{aligned} \tag{33}$$

Following triangle inequality, the distance between the combined action and the optimum action is as follows:

$$\begin{aligned}
|a_\beta(s) - a^\star(s)| &\geq |a_{\text{reg}}(s) - a^\star(s)| - |a_{\text{reg}}(s) - a_\beta(s)| \\
&= |a_{\text{reg}}(s) - a^\star(s)| - (1 - \beta(s))|a_{\text{reg}}(s) - a_{\text{rl}}(s)|.
\end{aligned} \tag{34}$$

To prove the convergence of $\pi_\beta$, we first examine the convergence of $\beta(s)$. According to Lemma 2, $\beta(s)$ converges to 0 under the assumption that the RL policy $\pi_{\text{rl}}$ converges to $\pi^\star(s)$ with $Q(s, \pi^\star) > Q(s, \pi), \forall \pi \neq \pi^\star$. Because $\beta(s) = 0$, the combined policy has variance equal to $\pi_\theta$. According to Lemma 1, the following holds for any state $s$:

$$\begin{aligned}
\mathbb{E}[a_\beta(s)] &= \underset{a}{\text{argmin}} \, \|a - \underset{a_{\text{rl}}(s)}{\text{argmax}} \, Q(s, a_{\text{rl}}(s))\|_\Sigma + \frac{\beta(s)}{1 - \beta(s)} \|a - a_{\text{reg}}(s)\|_\Sigma \\
&= \underset{a}{\text{argmin}} \, \|a - \underset{a_{\text{rl}}(s)}{\text{argmax}} \, Q(s, a_{\text{rl}}(s))\|_\Sigma = \bar{\pi}_{RL}(s).
\end{aligned} \tag{35}$$

Thus the the converged combined policy $\pi_\beta(s) = \pi_{\text{rl}}(s) = \pi^\star(s)$ with total variance distance $D_{\text{TV}}(\pi_\beta(s), \pi^\star(s)) = 0$. $\qquad\square$

## B Environments for validations

In this section, we introduce the four environments (*Glucose*, *BiGlucose*, *CSTR*, and *Cart Pole*) used in Section 4, with detailed environment models, parameters, and reward functions. MPC minimizes the stage cost $J_k$ and terminal cost $J_N$, while RL maximizes rewards $r$, for consistency, we set the MPC costs to $J_k = J_N = -r$ when validating the MPC in the environments.

### B.1 Glucose

Table 2: Glucose parameters for the estimated model and the actual environment

| Parameters | Unit | Estimated Model | Actual Environment |
|---|---|---|---|
| $G_b$ | mg/dL | 138 | 138 |
| $I_b$ | $\mu$U/mL | 7 | 7 |
| $n$ | min$^{-1}$ | 0.2814 | 0.2 |
| $p_1$ | min$^{-1}$ | 0 | 0. |
| $p_2$ | min$^{-1}$ | 0.0142 | 0.005 |
| $p_3$ | min$^{-1}$ | 15e-6 | 5e-6 |
| $D_0$ | - | 4 | 4 |
| $dt$ | min | 10 | 10 |

The Glucose environment simulates blood glucose level, denoted as $G$, against meal-induced disturbances which elevate $G$ and cause hyperglycemia. The observations are $(G, \dot{G}, t)$, where

$\dot{G} = G_t - G_{t-1}$ and $t$ is the total time passed after meal ingestion. The action is the injection of insulin $a_I$, which gradually lowers $G$ but with a large delay. Excessive injection of insulin can cause life-threatening hypoglycemia. Safety constraints are imposed on the blood glucose level $G$, as both the elevated and reduced $G$ result in severe health risks (hyperglycemia and hypoglycemia, respectively). The blood glucose model contains 3 state variables regulated by the ordinary differential equations (ODEs) given below:

$$\begin{aligned}
\dot{G} &= -p_1(G - G_b) - GX + D_t \\
\dot{X} &= -p_2 X + p_3(I - I_b) \\
\dot{I} &= -n(I - I_b) + a_I,
\end{aligned} \qquad (36)$$

among which only $G$ can be observed. Because not all states are observed, the closed loop is maintained only for the observed $G$ for MPC and RL-AR. The term $D_t$ represents the time-varying disturbance caused by the meal disturbance:

$$D_t = D_0 \exp(-0.01t). \qquad (37)$$

The model parameters adopted by the estimated model and the actual environment are given by Table 2. The initial states are determined by setting the left-hand side of Eq. (36) to zeros and solving the steady-state equations. The reward function for Glucose is the Magni risk function [Fox et al., 2020], which gives stronger penalties for low blood glucose levels to prevent hypoglycemia:

$$r = \begin{cases} -\left(3.35506 \times \left((\ln G)^{0.8353} - 3.7932\right)\right)^2, & 10 \leq G \leq 1000 \\ -1e5, & \text{otherwise} \end{cases}. \qquad (38)$$

### B.2 BiGlucose

Table 3: BiGlucose parameters for the estimated model and the actual environment

| Parameter | Unit | Estimated Model | Actual Environment |
|---|---|---|---|
| $D_G$ | kg | 0.08 | 0.08 |
| $V_G$ | L/kg | 0.14 | 0.18 |
| $k_{12}$ | $\text{min}^{-1}$ | 0.0968 | 0.0343 |
| $F_{01}$ | mmol/(kg min) | 0.0199 | 0.0121 |
| $EGP_0$ | mmol/(kg min) | 0.0213 | 0.0148 |
| $A_g$ | - | 0.8 | 0.8 |
| $tmax, G$ | min | 40 | 40 |
| $t_{max,I}$ | min | 55 | 55 |
| $V_I$ | L $\text{kg}^{-1}$ | 0.12 | 0.12 |
| $k_e$ | $\text{min}^{-1}$ | 0.138 | 0.138 |
| $k_{a1}$ | $\text{min}^{-1}$ | 0.0088 | 0.0031 |
| $k_{a2}$ | $\text{min}^{-1}$ | 0.0302 | 0.0752 |
| $k_{a3}$ | $\text{min}^{-1}$ | 0.0118 | 0.0472 |
| $k_{b1}$ | L/($\text{min}^2$ mU) | 7.58e-5 | 9.11e-6 |
| $k_{b2}$ | L/($\text{min}^2$ mU) | 1.42e-5 | 6.77e-6 |
| $k_{b3}$ | L/(min mU) | 8.5e-4 | 1.89e-3 |
| $t_{max,N}$ | min | 20.59 | 32.46 |
| $k_N$ | $\text{min}^{-1}$ | 0.735 | 0.620 |
| $V_N$ | mL $\text{kg}^{-1}$ | 23.46 | 16.06 |
| $p$ | $\text{min}^{-1}$ | 0.074 | 0.016 |
| $S_N \cdot 10^{-4}$ | mL/pg $\text{min}^{-1}$ | 1.98 | 1.96 |
| $M_g$ | g/mol | 180.16 | 180.16 |
| $BW$ | kg | 68.5 | 68.5 |
| $N_b$ | pg/mL | 48.13 | 48.13 |
| $dt$ | min | 10 | 10 |

The BiGlucose environment simulates blood glucose level $G$ against meal-induced disturbances, which elevate $G$ and cause hyperglycemia. The observations are $(G, \dot{G}, t)$, where $\dot{G} = G_t - G_{t-1}$

and $t$ is the total time passed after meal ingestion. The actions are insulin and glucagon injections $(a_I, a_N)$. Insulin injection $a_I$ lowers $G$ but causes hypoglycemia when overdosed. Glucagon injection $a_N$ elevates $G$ and thus can be used to mitigate the hypoglycemia caused by $a_I$. Similar to Glucose, safety constraints are imposed on $G$. The blood glucose model contains 12 internal states (11 of them unobservable) and 2 actions with large delays, regulated by the ODEs given below:

$$
\begin{aligned}
\dot{Q}_1 &= -F_{01}^c(G) - x_1 Q_1 + k_{12} Q_2 - F_R \\
&\quad + (1 - x_3)\text{EGP}_0 + c_{conv} U_G + Y Q_1 \\
\dot{Q}_2 &= x_1 Q_1 - (k_{12} + x_2) Q_2 \\
\dot{x}_1 &= -k_{a1} x_1 + k_{b1} I \\
\dot{x}_2 &= -k_{a2} x_2 + k_{b2} I \\
\dot{x}_3 &= -k_{a3} x_3 + k_{b3} I \\
\dot{S}_1 &= a_I - \frac{S_1}{t_{max,I}} \\
\dot{S}_2 &= \frac{S_1}{t_{max,I}} - \frac{S_2}{t_{max,I}} \\
\dot{I} &= \frac{S_2}{V_I t_{max,I}} - k_e I \\
\dot{Z}_1 &= a_N - \frac{Z_1}{t_{max,N}} \\
\dot{Z}_2 &= \frac{Z_1}{t_{max,N}} - \frac{Z_2}{t_{max,N}} \\
\dot{N} &= -k_N(N - N_b) + \frac{Z_2}{V_N t_{max,N}} \\
\dot{Y} &= -pY + pS_N(N - N_b),
\end{aligned}
\tag{39}
$$

where the intermediate variables $F_{01}^c$ and $F_R$ are piecewise functions of the measurable blood glucose mass $G = 18 \times Q_1/V_G$, as shown below:

$$
F_{01}^c = \begin{cases} F_{01}, & G \geq 81\text{mg/dL} \\ F_{01} G/81, & \text{otherwise} \end{cases},
\tag{40}
$$

$$
F_R = \begin{cases} 0.003(G/18 - 9)V_G, & G \geq 152\text{mg/dL} \\ 0, & \text{otherwise} \end{cases}.
\tag{41}
$$

Since only $G$ can be observed among the 12 states, the closed loop is maintained only for $G$ in MPC and RL-AR. The term $U_G$ represents the time-varying disturbance caused by the meal disturbance:

$$
U_G = \frac{D_G A_G}{t_{max,G}^2} \cdot t \cdot e^{-t/t_{max,G}}.
\tag{42}
$$

The model parameters adopted by the estimated model and the actual environment are given by Table 3. This extended model proposed by Herrero et al. [2013] and Kalisvaart et al. [2023] captures more complicated blood glucose dynamics, allowing the use of both insulin injection and glucagon injection as actions. This leads to better regulation performance, but also drastically increases the complexity of the problem due to its large number of unobservable states, delayed action responses, and nondifferentiable piecewise dynamics [Kalisvaart et al., 2023].

The initial states are determined by setting the left-hand side of Eq. (39) to zeros and solving the steady-state equations. The reward function for BiGlucose is the Magni risk function [Fox et al., 2020], which gives stronger penalties for low blood glucose levels to prevent hypoglycemia:

$$
r = \begin{cases} -10 \times \left(3.35506 \times ((\ln G)^{0.8353} - 3.7932)\right)^2, & 10 \leq G \leq 1000 \\ -1e5, & \text{otherwise} \end{cases}.
\tag{43}
$$

Table 4: CSTR actual environment model parameters

| Parameter | Unit | Actual Environment | Parameter | Unit | Actual Environment |
|---|---|---|---|---|---|
| $k_{0,ab}$ | h$^{-1}$ | 1.287e12 | $\rho$ | kg/L | 0.9342 |
| $k_{0,bc}$ | h$^{-1}$ | 1.287e12 | $C_p$ | kJ/kg.K | 3.01 |
| $k_{0,ad}$ | L/mol.h | 9.043e9 | $C_{p,k}$ | kJ/kg.K | 2.0 |
| $R_{gas}$ | kJ/mol.K | 8.3144621e-3 | $A_R$ | m$^2$ | 0.215 |
| $E_{A,ab}$ | kJ/mol | 9758.3 | $V_R$ | L | 10.01 |
| $E_{A,bc}$ | kJ/mol | 9758.3 | $m_k$ | kg | 5.0 |
| $E_{A,ad}$ | kJ/mol | 8560.0 | $T_{in}$ | °C | 130.0 |
| $H_{R,ab}$ | kJ/mol | 4.2 | $K_w$ | kJ/h m$^2$ K | 4032.0 |
| $H_{R,bc}$ | kJ/mol | -11.0 | $C_{A,0}$ | mol/L | 5.1 |
| $H_{R,ad}$ | kJ/mol | -41.85 | dt | h | 0.05 |

Table 5: CSTR different parameters for the estimated model and the actual environment

| Parameter | Estimated model | Actual Environment |
|---|---|---|
| $\alpha$ | 1 | 1.05 |
| $\beta$ | 1 | 1.1 |

## B.3 CSTR

The CSTR environment simulates the concentration of a target chemicals in a continuous stirred tank reactor. The observations are $(C_A, C_B, T_R, T_K)$, where $C_A$ and $C_B$ are the concentrations of two chemicals, $T_R$ is the temperature of the reactor, and $T_K$ is the temperatures of the cooling jacket. The actions are the feed and the heat flow $(a_F, a_Q)$. Safety constraints are imposed on the chemical concentrations and reactor temperature, as crossing the safe boundaries for any of them can lead to tank failure or even explosions. This model contains four state variables regulated by the ODEs given below:

$$\dot{C_A} = a_F(C_{A,0} - C_A) - K_1 C_A - K_3 C_A^2$$
$$\dot{C_B} = -a_F C_B + K_1 C_A - K_2 C_B$$
$$\dot{T_R} = \frac{K_1 C_A H_{R,ab} + K_2 C_B H_{R,bc} + K_3 C_A^2 H_{R,ad}}{-\rho C_p} + \frac{K_w A_R (T_K - T_R)}{\rho C_p V_R} + a_F(T_{in} - T_R) \quad (44)$$
$$\dot{T_K} = \frac{a_Q + K_w A_R (T_R - T_K)}{m_k C_{p,k}},$$

where the intermediate variables are:

$$K_1 = = \beta k_{0,ab} \exp(\frac{-E_{A,ab}}{T_R + 273.15})$$
$$K_2 = = k_{0,bc} \exp(\frac{-E_{A,bc}}{T_R + 273.15}) \quad (45)$$
$$K_3 = = k_{0,ad} \exp(\frac{-\alpha E_{A,bc}}{T_R + 273.15}).$$

The model parameters adopted by the estimated model and the actual environment are given by Table 4 and Table 5. The initial concentration of the target chemical $C_{B,0}$ is set to 0.5. The reward function for CSTR is as follows:

$$r = \begin{cases} -(100 \times (C_B - 0.6)^2, & 0.1 \leq C_A \leq 2,\ 0.1 \leq C_B \leq 2,\ 50 \leq T_R \leq 200,\ 50 \leq T_K \leq 150 \\ -(100 \times (C_B - 0.6)^2 - 1e4, & \text{otherwise} \end{cases}. \quad (46)$$

## B.4 Cart Pole

The Cart Pole environment simulates an inverted pole on a cart. The environment is the continuous action adaptation of the gymnasium environment [Towers et al., 2023]. The observations are

Table 6: Cart Pole parameters for the estimated model and the actual environment

| PARAMETER | UNIT | ESTIMATED MODEL | ACTUAL ENVIRONMENT |
|---|---|---|---|
| $g$ | $m \cdot s^{-2}$ | 9.8 | 9.8 |
| $m_c$ | kg | 1.0 | 0.8 |
| $m_p$ | kg | 0.1 | 0.3 |
| $l$ | m | 0.5 | 0.6 |
| $dt$ | s | 0.02 | 0.02 |

Table 7: RL-AR hyperparameters. The baseline methods utilized the same network structures and training hyperparameters.

| Parameter | Value |
|---|---|
| Learning rate for Q network | $1 \times 10^{-3}$ |
| Learning rate for policy network | $3 \times 10^{-4}$ |
| Batch size $|\mathcal{B}|$ for updating | 256 |
| Start learning | 256 |
| Target Q network update factor $\tau$ | 0.005 |
| Forgetting factor $\gamma$ | 0.99 |
| Frequency for updating policy network | 2 |
| Frequency for updating target network | 1 |
| Learning rate for the focus module | $5 \times 10^{-6}$ |
| focus module pretraining threshold $1 - \epsilon$ | 0.999 |
| Minimum log policy variance | $-5$ |
| Maximum log policy variance | 2 |
| Policy network hidden layers | $[256, 256]$ |
| Q Network hidden layers | $[256, 256]$ |
| focus module hidden layers | $[128, 32]$ |
| Glucose MPC horizon | 100 |
| Other envs MPC horizon | 20 |

$(x, \dot{x}, \theta, \dot{\theta})$, where $x$ is the position of the cart and $\theta$ is the angle of the pole. The action is the horizontal force $a_f$. Safety constraints are imposed on $x$ and $\theta$ as the control fails if the cart reaches the end of its rail or the pole falls over. This model contains four state variables regulated by the ODEs given below:

$$\ddot{\theta} = \frac{g \sin \theta - d \cos \theta}{l(4/3 - m_p \cos^2 \theta / (m_p + m_c))}$$
$$\ddot{x} = d - \frac{m_p l \ddot{\theta} \cos \theta}{m_p + m_c} \tag{47}$$

The intermediate variable $d$ is:

$$d = \frac{10 \times a_f + m_p l \dot{\theta}^2 \sin \theta}{m_p + m_c} \tag{48}$$

The model parameters adopted by the estimated model and the actual environment are given by Table 6. The initial tilt of the pole is 6 degrees. The reward function for Cart Pole is as follows:

$$r = \begin{cases} -1000\theta^2 - \max(0, |x| - 0.25), & -2.4 \le x \le 2.4, \ -12\pi/360 \le \theta \le 12\pi/360 \\ -1000\theta^2 - \max(0, |x| - 0.25) - 1e4, & \text{otherwise} \end{cases} . \tag{49}$$

## C  Implementation details

Experiments are conducted using Python 3.12.5 on an Ubuntu 22.04 machine with 13th Gen Intel Core i7-13850HX CPU, Nvidia RTX 3500 Ada GPU, and 32GB RAM. For RL-AR, the average time for taking a step (interaction and network updates) is 0.0235 s for Glucose, 0.0667 s for BiGlucose,

0.0378 s for CSTR, and 0.0206 s for Cart Pole. The above decision times are practical for real-time control in these environments.

The MPC and the safety regularizer in RL-AR are implemented using [Fiedler et al., 2023]. The implementations for SAC and the RL agent in RL-AR are based on [Huang et al., 2022]. RPL, CPO, and SEditor follow the implementations in [Silver et al., 2014], [Ray et al., 2019], and [Yu et al., 2022b], respectively. The same hyperparameters are used by RL-AR for all experiments in Section 4, which are listed in Table 7. The baseline methods' network structure and training parameters are set to be the same as RL-AR.

## D   Additional experiment results

### D.1   State-dependent vs. scalar policy combination

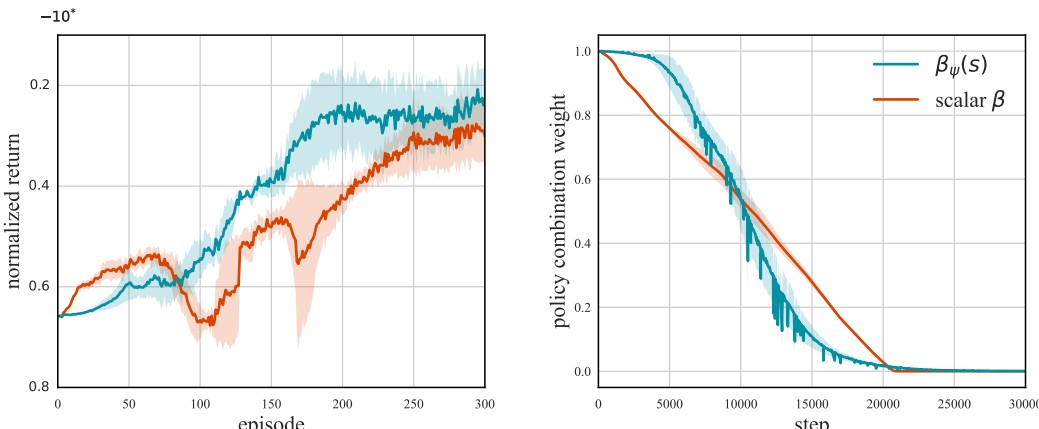

Figure 5: Comparing the state-dependent focus module $\beta_\psi(s)$ with the scalar $\beta$ by plotting the normalized return curves (left) and focus weight curves (right) in the Glucose environment. Shaded areas indicate standard deviations.

As discussed in Section 3 and Theorem 2, using a state-dependent focus module $\beta(s)$ for policy combination (compared to using a scalar weight $\beta$) offers the advantage of achieving monotonic performance improvements at least in the tabular setting. Here, we empirically verify the advantage of using the state-dependent $\beta_\psi(s)$ versus a scalar weight $\beta$ in Fig. 5 by analyzing the normalized returns (left panel) and the focus weights (right panel) during training in the Glucose environment (Fig. 5). The mean (solid lines) and standard deviation (shaded area) in Fig. 5 are obtained from 5 independent runs using different random seeds.

Practically, RL-AR with state-dependent $\beta_\psi(s)$ does not show a strictly monotonic policy improvement, which can be attributed to the neural network approximation. However, improvements in the normalized return are significantly more steady when using $\beta_\psi(s)$ rather than a scalar $\beta$, as shown by the blue and red curves in the left panel of Fig. 5. The right panel of Fig. 5 shows the evolution of the focus weights (used by RL-AR) versus the training steps. Although both $\beta_\psi(s)$ and $\beta$ converge to zero after approximately the same number of steps, $\beta_\psi(s)$ applies different focus weights depending on specific states encountered as seen by the fluctuations in the blue curve in the right panel of Fig. 5.

### D.2   Entropy Regularization

We conduct an ablation study to compare using SAC as the RL agent in RL-AR with using another state-of-the-art RL algorithm, TD3 [Fujimoto et al., 2018]. The key difference between SAC and TD3 is that SAC incorporates the entropy regularization term, $-\alpha \log P_{\pi_\theta}(a|s)$, in Eq. (3) and Eq. (5). The normalized return curves, shown in Fig. 6, demonstrate that RL-AR with SAC as the RL agent achieves a higher normalized return at a faster rate. While RL-AR promotes safety and stability at the cost of reducing the exploration intensity of the combined policy (as proved in Lemma 1), which could potentially slow down the discovery of the optimal policy, SAC's entropy regularization

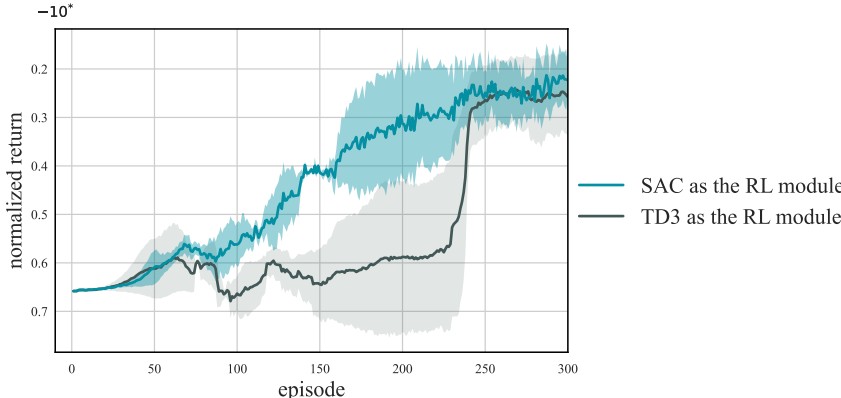

Figure 6: Comparison of normalized return between using the SAC and using TD3 [Fujimoto et al., 2018] as the RL agent in the Glucose environment (standard deviations are shown in the shaded area). The main difference between SAC and TD3 is that SAC has the entropy regularization terms in its objectives, which are intended to encourage diverse policies and stabilize training.

counteracts this by promoting the use of more diverse policies. A closer examination reveals that the SAC curve locally exhibits more minor fluctuations than the TD3 curve, illustrating SAC's ability to use more diverse policies.

### D.3 focus weight curves during training

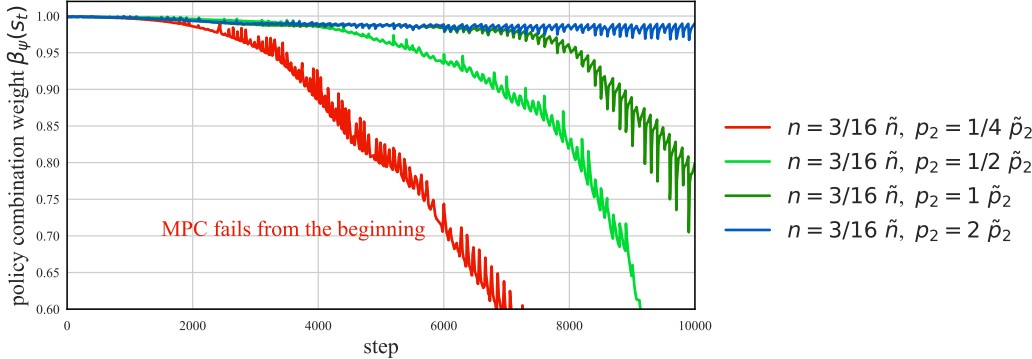

Figure 7: The focus weights when training with varying levels of discrepancies between the estimated Glucose model (with parameters $\tilde{p}_2, \tilde{n}$) and the actual Glucose environment (with parameters $p_2, n$).

In Fig. 7, we show several $\beta_\psi(s_t)$ curves from training RL-AR in various Glucose environments, created by varying the environment model parameters $n$ and $p_2$. Let $\tilde{n}$ and $\tilde{p}_2$ be the parameters of the estimated model $\tilde{f}$, the actual environment models have $n = 3\tilde{n}/16$ and $p_2 = \tilde{p}_2/4, \tilde{p}_2/2, 1\tilde{p}_2, 2\tilde{p}_2$ to mimic deviating characteristics of new patients. When there are large discrepancies between the environment and $\tilde{f}$ (e.g., $n = 3\tilde{n}/16$ and $p_2 = \tilde{p}_2/4$), the safety regularizer fails initially, but the focus weight $\beta_\psi(s_t)$ decreases rapidly, enabling RL-AR to recover from initial failures by rapidly shifting from the sub-optimal safety regularizer policy to the stronger learned RL policy. Conversely, $\beta_\psi(s_t)$ converges more slowly to zero when the safety regularizer performs well in the actual environment, as it becomes more challenging to find an RL policy that significantly outperforms the safety regularizer policy in such cases.

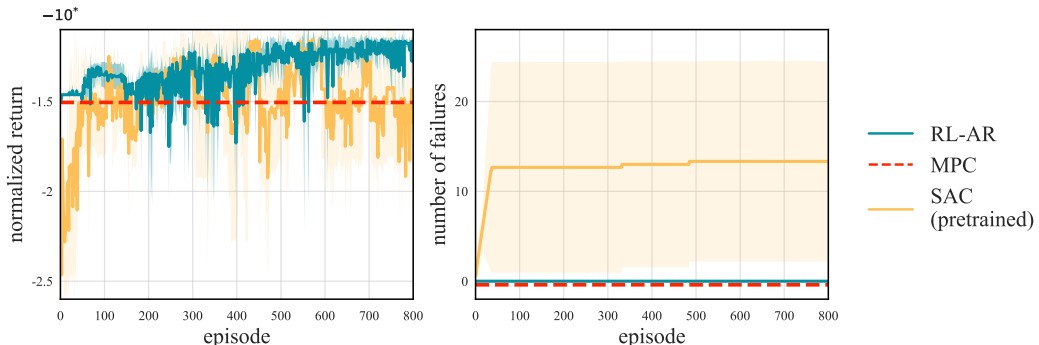

Figure 8: Normalized return (left) and the number of failures (right) during training in the Acrobot environment (standard deviations are shown in the shaded area).

## D.4    The Acrobot environment

In Section 4, we evaluate RL-AR in several widely recognized challenging safety-critical environments. For example, the BiGlucose environment (see Appendix B.2) involves 11 unobservable states, two actions with significant delays, and complex, nondifferentiable piecewise dynamics. Here, we provide further evidence of the effectiveness of RL-AR in the Acrobot environment, an adaptation of the Gymnasium environment [Towers et al., 2023] with a continuous action space (Fig. 8). The Acrobot environment simulates two links connected by a joint, with one end of the connected links fixed. The links start facing downward. The objective is to swing the free end above a given target height as quickly as possible by applying torque to the joint. Failure is defined as the tip not reaching the target height in 400 time steps. For validation, we set the actual Acrobot environment parameters $l_2 = 1.1\tilde{l}_2, m_2 = \tilde{m}_2$, where $\tilde{l}_2 = 1.0$ is the lower link length parameter of the estimated model $\tilde{f}$, and $\tilde{m}_2 = 1.0$ is the lower link weight parameter of the estimated model $\tilde{f}$.

The Acrobot environment is particularly challenging for RL-AR's policy regularizer due to: i) its highly nonlinear, under-actuated dynamics, and ii) its definition of failure as not achieving the target within a given time limit, which cannot be easily formalized as a constraint. As Fig. 8 shows, RL-AR can swing up the tip in the first episode by initially relying on the viable safety regularizer policy. Throughout training, RL-AR ensures safety while converging to a similar normalized return as SAC, which focuses only on return and fails many times during training. This illustrates RL-AR's robustness in challenging tasks and its potential in applications where failures are defined in terms of time limit and are hard to formalize as constraints.

## E    Impact statement

This paper presents work that aims to advance the field of Machine Learning. There are many potential societal consequences of our work, none of which, based on our judgment, must be specifically highlighted here.

