# OpenReview forum: "Reinforcement Learning with Adaptive Regularization for Safe Control of Critical Systems"
_NeurIPS.cc/2024/Conference — NeurIPS 2024 poster_

### Official Review · Reviewer_zn6h · 2024-06-16

**Soundness:** 3
**Presentation:** 4
**Contribution:** 3
**Rating:** 6
**Confidence:** 4

**Summary:**

The authors develop RL with Adaptive Control Regularization (RL-ACR) to enable safe RL exploration. This solution uses two agents: a safety regularizer to enforce safety constraints, and an adaptive agent to perform exploration. They demonstrate their method on four critical applications against leading benchmarks.

**Strengths:**

1. The use of two parallel agents (safety regularizer and adaptive) is novel.

2. The work is clearly presented with good explanations, algorithm procedures, and visuals.

3. The work studies real-world application problems.

4. The evaluations test the proposed RL-ACR method against a large number of baselines.

5. The evaluations substantiate the authors' claims of 1) improved performance, and 2) reduced failure rate.

6. The evaluations are accompanied by thorough analysis and discussion of the results by the authors.

**Weaknesses:**

1. Theorem 1 requires the assumption that the RL policy converges to the optimal policy. However, I do not see a result guaranteeing that the RL-ACR policy converges to the optimal. Instead, I see a monotonic performance improvement result in Theorem 2, but this is not an optimality result.

2. The work is intended to address safety, yet the authors do not provide a theoretical guarantee of closed-loop stability. Such guarantees are important for control of dynamical systems (e.g., airplanes, cars).

**Questions:**

1. If an optimality result is available, does it require that the estimated system model $\tilde{f}$ be exact to the actual physical process? I am thinking about this in light of your assumptions for Theorem 1. Presumably an imperfect model would void the convergence to the optimal policy and thereby the sufficient conditions required for Theorem 1. I do appreciate the empirical attention paid to model uncertainty in Section 4.2.

2. How do the compute/data requirements of RL-ACR compare to the baselines?

**Limitations:**

The authors adequately address limitations.

---

> ### Author Rebuttal · Authors · 2024-08-07
>
> **[Weakness 1]**: The reviewer asked about the convergence result of RL-ACR.
>
> **Response**: For state $s$, the regulation on RL policy converges to 0 ($\beta(s)$ converges to 0) with the assumption that RL learns a better policy than the regularizer. This is a reasonable assumption since the regularizer was derived from a suboptimal model. When the regularization converges to 0, the problem is equivalent to a standard RL problem, where the assumption of RL convergence is reasonable and widely accepted in research. We mentioned this in the proof in L487-492, but we thank the reviewer for the review and will clarify the assumption for Theorem 1.
>
> **[W2]**: The reviewer asked about the closed-loop stability of RL-ACR.
>
> **Response**: We fully understand the reviewer’s point but would like to add a couple of remarks that might help their assessment. We considered not knowing the exact model (a characteristic phenomenon in real-world application) and thus did not focus on finding a theoretical guarantee of closed-loop stability, which would be model-dependent, and the analysis would be application-specific. However, we believe that the proof on guaranteed policy return improvement (although addressing the tabular case) is a valuable theoretical result for RL-based methods such as ours.
>
> **[Question 1]**: The reviewer asked if an imperfect model prevents the convergence to the optimality result.
>
> **Answer**: The optimality proof does not depend on the perfect model. In fact, we assume that the model used for MPC is suboptimum. If the model is imperfect, stronger policies than MPC exist and can be explored by the RL policy. This causes $\beta(s)$ to converge to 0. If $\beta(s)$ converges to 0, there is no regularization term for the policy optimization, thus reducing the problem to a standard RL problem with proven convergence to the optimum policy.
>
> **[Q2]**: The reviewer asked about the compute/data requirements of RL-ACR compared to the baselines.
>
> **Answer**: the average per step solving time for RL-ACR is 0.0375 on one CPU, which is similar to MPC (widely adopted in applications). Model-free methods such as SAC and CPO are faster than RL-ACR, but they cause constraint violations during training. Considering the RL-ACR solve time is sufficient for real-time control, the computation requirement is reasonable to guarantee safety during training. In terms of data, RL-ACR and the baselines collect the same amount of environment samples.

---

> > ### Comment · Reviewer_zn6h · 2024-08-09
> > **My thanks to the authors. Overall, I remain a positive review of this work. However, I am troubled by assertions made in the rebuttal, which does not convincingly address the raised weeknesses. Decrementing my score.**
> >
> > My thanks to the authors for their responses to my posed weaknesses and questions.
> >
> > The positive aspects of this paper I identified in the review remain unchanged.
> >
> > However, the authors failed to adequately address my riased weaknesses. In fact, their response, which borders on cavalier at times, raises concerns I did not have previoulsy.
> >
> > In light of these points, I am decrementing my score to a weak accept.
> >
> > Below can be found my responses to your specific points. I have focused on the remarks which troubled me:
> >
> > **[Weakness 1]:**
> >
> > > For state $s$ the regulation on RL policy converges to 0 ($\beta(s)$ converges to 0) with the assumption that RL learns a better policy than the regularizer.
> >
> > * I am confused by the point the authors are trying to make here. $\beta(s)$ converging to zero implies that the focus is choosing the RL module agent you are presenting to it -- it says nothing about the convergence of the RL module policy or its quality.
> >
> > * **"Better"** here is vaguely defined and does not imply **convergence** at all.
> >
> > * My weakness centers on the failure of the convergence of the **sequence of policies** (i.e., convergence with respect to the algorithm iteration of the training scheme) to the optimal policy, not on the focus network simply selecting between the safety regularizer and the RL module.
> >
> > > [...] with the assumption that RL learns a better policy than the regularizer.
> >
> > * This is not rigorous or substantial by any means. It's expected that your focus network should pick the RL module policy **if it's assumed to be a better policy.** I'm asking for **justification via provable results that the RL module converges and delivers improved performance.**
> >
> > > [...] the assumption of RL convergence is reasonable and widely accepted in research
> >
> > * This comment is very troubling. In other words, you mean to assert that every RL algorithm developed is universally accepted to converge?
> >
> > * So then, **simply calling your algorithm an RL algorithm automatically gives it established convergence performance?**
> >
> > * I don't see any other way to interpret this statement.
> >
> > **[Weakness 2]:**
> >
> > > a theoretical guarantee of closed-loop stability [...] would be model-dependent, and the analysis would be application-specific.
> >
> > * This is simply not true in control settings -- in fact, the vast bulk of standard stability results in the control literature are formulated for general systems rather than application-specific examples.
> >
> > > We considered not knowing the exact model (a characteristic phenomenon in real-world application) and thus did not focus on finding a theoretical guarantee of closed-loop stability,
> >
> > * This seems to suggest that model uncertainty and stability guarantees are mutually exclusive, which is also not the case in the literature at all.
> >
> > * In fact, there is a really important subtlety here: One of the core reasons for opting for RL frameworks is to address uncertainty in the environment. The authors seem to suggest that performance guarantees cannot be derived in the presence of model uncertainty -- exactly the core use-case of RL.

---

> > > ### Author Response · Authors · 2024-08-09
> > > **Our responses and actions regarding the weaknesses pointed out by the reviewer (as well as the reviewer's concerns)**
> > >
> > > We thank the reviewer for the feedback. Our response was intended to render our agreement with the Weakness points identified by the reviewer whilst raising additional points that may aid the assessment of our work; we sincerely apologize if the response seemed dismissive in any form. Below, we provide clarifications for our responses (taking the reviewer’s concerns into account) as well as our action on the manuscript to address the received feedback.
> > >
> > > > **[Weakness 1]:**
> > >
> > > The reviewer correctly identified that: Theorem 1 requires the assumption that the RL policy converges to the optimal policy, and that, this policy converging to the optimal is not established anywhere in the manuscript. This assumption, however, is justified by an existing proof in [Haarnoja et al., 2018] and we will revise the manuscript appropriately to address this comment.
> > >
> > > More specifically, our method uses the SAC algorithm in its RL module, which has been proven to “converge to the optimum policy” before in [Haarnoja et al., 2018]. Our Theorem 1 takes this existing result (RL module convergence) as an assumption and provides more insight into the behavior of the combined policy. We will revise Theorem 1 in the manuscript, and in particular, we will add the following to the paper, directly referring to the existing proof of RL module convergence that we take as an assumption for further analysis.
> > >
> > > “[…] Haarnoja et al. [2018] prove (see Theorem 1 in the reference) that the learned RL policy with the Soft Actor-Crtic losses formulated in Eqs. (3) and (6), after repeated updates converges to a policy $\pi^\star$, such that no policy with a larger expected return than that of $\pi^\star$ can be found, i.e., $Q^{\pi^\star}(s_t, a_t) \geq Q^{\pi }(s_t, a_t)$ for all $\pi$ and $(s_t, a_t)$.”
> > >
> > > We would like to add that by “a better policy” in our response, we meant a policy with a higher expected return, and we extend our apology for using a vague/inaccurate term only to compress our response. We hope that the reviewer finds the updated response/action appropriate for the weakness pointed out.
> > >
> > > > **[Weakness 2]:**
> > >
> > > We agree with the reviewer’s both initial and the new comment on [Weakness 2]. Although we completely agree with its significance here, a theoretical closed-loop stability guarantee is very difficult to achieve where RL (since the agent is blind to the model) is involved. Therefore, we took the approach of demonstrating that the policy update will only improve or retain the expected return of the combined policy. This indicates that the proposed model is guaranteed (proved in Theorem 2 for the tabular case) to perform at least as well as (again in terms of the expected return) the initial regularizer module. We will clarify the above further in Theorem 2 and will state the weakness pointed out by the reviewer in the Conclusion section as a limitation of the approach.
> > >
> > > We have made an unintentional error in the rebuttal, which we understand sent an unclear message. We agree with the reviewer that model uncertainty and stability guarantees are not mutually exclusive, and our performance guarantee is simply approached from a policy improvement perspective rather than a closed-loop stability perspective.

---

> > > > ### Comment · Reviewer_zn6h · 2024-08-09
> > > > **Thanks to the authors for clarifying these points. Retaining my scores.**
> > > >
> > > > **Weakness 1.**
> > > >
> > > > * Overall, I appreciate the clarification provided. Please include the appropriate citation.
> > > >
> > > > > More specifically, our method uses the SAC algorithm in its RL module
> > > >
> > > > * I am trying to follow the logic here:
> > > >
> > > > 1. The authors provide a proof that the regularizer $\beta$ chooses the RL Module policy if it exhibits improved performance.
> > > >
> > > > 2. The RL module is SAC.
> > > >
> > > > So the end result here is that your converged policy is simply SAC?
> > > >
> > > > * We therefore can't avoid the following conclusion: **your final converged policy can perform no better than SAC (a standard method).**
> > > >
> > > > * Why not bypass the extra complexity of your regularizer and choose to implement SAC directly then, if SAC already comes with a stability guarantee?
> > > >
> > > > > We would like to add that by “a better policy” in our response, we meant a policy with a higher expected return [...]
> > > >
> > > > * I understood the authors's intent, but I want to point out that they have simply offloaded the vagueness from **"better"** to **"higher"**. This is still not quantitative/systematic.
> > > >
> > > > * With this said, I now indirectly have somewhat of a foothold to establish some quantitative guarantees from the method, which it inherits from SAC. But this runs into the issues I delineate in my above points.
> > > >
> > > > **Weakness 2.**
> > > >
> > > > > This indicates that the proposed model is guaranteed (proved in Theorem 2 for the tabular case) to perform at least as well as (again in terms of the expected return) the initial regularizer module.
> > > >
> > > > * I see the point made. But the final conclusion which must be made is that the method achieves:
> > > >
> > > > 1. Performance no worse than the regularizer (MPC).
> > > >
> > > > 2. Performance no better than the RL module (SAC).
> > > >
> > > > So the results deliverable sit between standard methods already available.

---

> > > > > ### Comment · Reviewer_zn6h · 2024-08-09
> > > > > **One last clarifying remark in this discussion.**
> > > > >
> > > > > And I want to make it clear here: I recognize the novelty of **the idea of weighting two independent policies** as providing novel elements to this work (see my original review), rather than strictly basing my score on **the standard policies which have been combined**
> > > > >
> > > > > If you will, my score lies in a convex combination of these two points -- my own focus network of sorts. I remain at a weak accept and leave further deliberation to the ACs.

---

> > > > > ### Author Response · Authors · 2024-08-10
> > > > >
> > > > > We sincerely thank the reviewer for the feedback and respect the decision; we have revised the manuscript to reflect the feedback as detailed in our previous response. Here, we add a few minor clarifications in response to the most recent comments for your consideration. For clarity, first, we would like to state that this manuscript deals with two performance aspects: i) the expected cumulative reward (i.e. the expected return) measured by Eq. (1) in the manuscript, ii) safety measured by the number of failures (defined as breaking the safety bounds).
> > > > >
> > > > > > Reviewer (Re W1): …your final converged policy can perform no better than SAC (a standard method). Why not bypass the extra complexity of your regularizer and choose to implement SAC directly then, if SAC already comes with a stability guarantee?
> > > > >
> > > > > **Response**: That is theoretically correct only in terms of “the cumulative reward”. However, our policy remains significantly safer (in terms of the number of failures) all the way from initiation of training to convergence when compared to the standard SAC. Bypassing the regularizer would harm the method’s performance substantially in terms of safety.
> > > > >
> > > > > > Reviewer (Re W1): [Authors] have simply offloaded the vagueness from "better" to "higher". This is still not quantitative/systematic.
> > > > >
> > > > > **Response**: “Higher expected return” (i.e., the expected cumulative reward) is quantified by Eq. (1) of the paper. This is what we referred to in our previous response and we hope it is clarified further now.
> > > > >
> > > > > > Reviewer (Re W2): I see the point made. But the final conclusion which must be made is that the method achieves: Performance no worse than the regularizer (MPC). Performance no better than the RL module (SAC).
> > > > >
> > > > > **Response**: The reviewer is correct with this conclusion; note that this is a theoretical result and for performance in terms of the “expected cumulative reward”. Practically, the proposed RL-ACR trains with safe actions (due to the effect of the regularizer) unlike the standard SAC, so the conclusion (more optimistically if you will) would be that the proposed method performs up to SAC in terms of “cumulative reward” and performs significantly better in terms of safety (measured by the number of failures). (Note that SAC achieves the state-of-the-art cumulative reward.)

---

> > > > > > ### Comment · Reviewer_zn6h · 2024-08-10
> > > > > > **Thanks to the authors**
> > > > > >
> > > > > > My thanks to the authors for this discussion and for these concluding points of clarification.

---

### Official Review · Reviewer_QQKo · 2024-06-23

**Soundness:** 1
**Presentation:** 4
**Contribution:** 1
**Rating:** 3
**Confidence:** 4

**Summary:**

The paper proposed a safe RL algorithm using adaptive regularization from model predictive controller. The regularization is implemented via weighted sum of a MPC controller and model-free RL policies. The weights is adaptive and learnable through a “focus network”. The focus network is updated through optimizing the expected Q function of weighted sum of actions. Experimental results shows that the proposed RL algorithm can learn a safe policy with zero constraint violations and outperforms MPC on the performance.

**Strengths:**

The idea is interesting to combine MPC as regularizers. The writing is clear and easy to understand. The experimental results looks promising under strict assumptions of known models.

**Weaknesses:**

1. There is a key assumption that, the MPC policy $\pi_r$ via estimated model is safe, is too strict, not realistic and not mentioned explicitly. The authors claimed that the model is sub-optimal in line 31, which means the model is not accurate. The model mismatches usually cause constraint violations, but the experimental results in Table 1 showed no constraint violations for MPC policies.
2. The computational cost is too high, which makes the proposed algorithm not generalizable to nonlinear/high-dimensional inputs/ unknown models. Solving a MPC is very computationally heavy. The authors didn’t mention what optimization package, how many CPUs they used for solving MPC problem (7). For nonlinear system, MPC are not guaranteed to be solved in polynomial time and converged to global optimal solutions. For system with visual inputs and unknown models, the methods are not applicable.
3. The focus network update might lead to unsafe policy.

**Questions:**

1. Can you clarify what types of references model $\tilde f$ you assumed for the regularizer policy $\pi_r$? See weakness 1 for the reason.
2. Why not compared to the Cheng et, al. as you already mentioned it? Now you only has one safe RL baseline CPO (and its variants). CPO is too old.
3. If you already has the safe model, why not just do CBF-QP on your learned policy? What is the potential advantages of the proposed algorithm
4. Please clarify the MPC solver and computational resources used for solving MPC.
5. For the focus network update, it seems like the focus network will finally choose $\beta(s) = 0$, which is the optimal policy. But the optimal policy might not be safe. What guarantees the safe policy is learned as you shown in Table 1?

[1] Richard Cheng, Gábor Orosz, Richard M Murray, and Joel W Burdick. End-to-end safe reinforcement learning through barrier functions for safety-critical continuous control tasks. In Proceedings of the AAAI conference on artificial intelligence, volume 33, pages 3387–3395, 2019a.
[2] Ames A D, Coogan S, Egerstedt M, et al. Control barrier functions: Theory and applications[C]//2019 18th European control conference (ECC). IEEE, 2019: 3420-3431.

**Limitations:**

The authors addressed the limitation adequately.

---

> ### Author Rebuttal · Authors · 2024-08-07
>
> **[Weakness 1]**: The reviewer stated that assuming the existence of a safe MPC policy is too strict, not realistic, and not mentioned in the manuscript.
>
> **Response**: we believe that the reviewer may have missed a few points. First, this assumption is explicitly mentioned and discussed in the paper at **L335**. Second, the existence of an estimated model for MPC to generate a safe policy is realistic and reasonable, supported by the fact that MPC is arguably the most widespread method used for constrained control applications. Furthermore, to showcase the amount of uncertainty that our approach can withstand before failing, we test the sensitivity of RL-ACR with respect to the model accuracy. This is shown in **Fig. 4**, which shows that RL-ACR maintains safety even when model parameters have > 60% deviations.
>
> **[W2]**: The reviewer stated that we have not mentioned the optimization package, number of CPUs, etc. used for our experiments and that the high computational cost of MPC is a weakness of our work. Moreover, they state that the proposed approach is not applicable to systems with unknown models and visual inputs.
>
> **Response**: In **L560**, we reported that the average decision+learning time for RL-ACR is 0.037s on a single CPU, which is reasonable for most real-time control applications. It is explicitly mentioned in  **L122** that we use do-mpc with the IPOPT solver package. The solver reduces the computational complexity by using the solution from the previous time step as the initial guess, leveraging the fact that MPC solves similar problems with slight variations at each time step. Moreover, while we agree with the reviewer that RL-ACR is not readily applicable to systems with unknown models and visual inputs, we want to stress two points:
>
> * Having an approximate model of the system at hand is the starting point of our work. This is a realistic assumption for many applications in real-world scenarios, where approximated models can be obtained by data-driven methods.
>
> * A very large class of problems does not rely on visual inputs. While RL-ACR could not be directly applied to systems that only rely on visual inputs (e.g., the ATARI suite), it can be applied to numerous other domains (e.g., medical applications, robotics, chemical processes, industrial settings).
>
> **[W3]**: The reviewer stated that the focus network update might lead to unsafe policy.
>
> **Response**: The focus network is iteratively updated using gradient ascent, which prevents $\beta(s)$ (the state-dependent focus weight) from a sudden decrease. This ensures regularization even if the learned RL policy is affected by overestimation error. The regularization from the safe policy is reduced to a small degree only after state $s$ is visited multiple times.
>
> **[Question 1]** The reviewer asked for clarification of reference(estimated) model used.
>
> **Answer**: The MPC reference models are widely adopted for safety-critical control applications [1]. In our work, the model parameters used to obtain the safe policy regularizer for Glucose and BiGlucose are estimated with real patient data, just as it can be practically done in real-world scenarios [2,3]. The environment models with their actual and estimated parameters are detailed in **Appendix B**.
>
> **[Q2]**: The reviewer suggested that we should have compared with an additional method (Cheng et al) and that our baseline (CPO) is too old.
>
> **Answer**: CPO is a well-established baseline used by most of the similar works in the recent literature (to give an example see the very recent paper [4]). Nevertheless, we compared our method to two additional more up-to-date baselines (Silver et al., 2018) and (Yu et al., 2022) in Figure R.I in the PDF attached with the global rebuttal. The results show that RL-ACR outperforms the new baselines, providing more evidence of the effectiveness of the proposed algorithm for safe control.
>
> **[Q3]**: The reviewer asked for the advantages of our method over CBF-QP.
>
> **Answer**: CBF-QP minimizes the control effort, without considering performance. The advantage of MPC is that it optimizes the predicted performance while maintaining safety constraints, resulting in a high performance of RL-ACR during the learning phase. Also, note that the theoretical safety guarantee of CBF-QP relies on the existence of a “perfect” model.
>
> **[Q4]**: The reviewer asked for clarification on the MPC solver and computation resources used.
>
> **Answer**: The original manuscript provides this information. MPC was solved using do-mpc, with its implementation based on CasADi and IPOPT (**L122** and **L197**). The average decision+learning time for RL-ACR is 0.037s on a single Intel Core i7-13850HX CPU (**L-560**).
>
> **[Q5]**: The reviewer raised a concern that after \beta(s) reaches 0 the proposed method might not be safe.
>
> **Answer**: In the problems, the policy that optimizes return is within the safe region by design. The state-dependent $\beta(s)$ will converge to 0 only for states where a better policy than the safe regularizer policy has been learned. If the exploration continues even after converging to the optimum policy and the system runs into less-exploited states,  $\beta(s)$ will be non-zero and RL-ACR will apply safe regularization.
>
> $~$
>
> [1] Sherr, Jennifer L., et al. "ISPAD clinical practice consensus guidelines 2022: diabetes technologies: insulin delivery." Pediatric diabetes 23.8, 2022.
>
> [2] Zahedifar, Rasoul, and Ali Keymasi Khalaji. "Control of blood glucose induced by meals for type-1 diabetics using an adaptive backstepping algorithm." Scientific Reports, 2022.
>
> [3] Hovorka, Roman, et al. "Nonlinear model predictive control of glucose concentration in subjects with type 1 diabetes." Physiological measurement, 2004.
>
> [4] Anderson G, Chaudhuri S, Dillig I. Guiding Safe Exploration with Weakest Preconditions. In The Eleventh International Conference on Learning Representations, 2023.

---

> ### Comment · Reviewer_QQKo · 2024-08-09
>
> Thanks for the detailed rebuttal.
>
> ### Re to response to W1:
>
> First, L335 is the limitation section, you mentioned here does not make the assumption more reasonable. For the results in figure 4, parameter uncertainty is only one of many possible reasons for the model uncertainty. A safe MPC controller for uncertain dynamics is generally very difficult.
>
> ### Re to response to W2.
>
> Thanks for the clarification on the calculation time. I won’t call it fast (you might need to consider the computational limit on the real hardware, not all real-time control application has a CPU as powerful as an intel i7.) but it’s okay.
>
> ### Re to response to W3:
>
> What I mean is that you have a linear combination of MPC actions and RL actions. These two actions might be all safe, but there is no guarantee that the intermediate actions are safe, which is the key weakness of this paper.
>
> ### Re to response to Q1 and Q2, Q4:
>
> my concerns are addressed. thanks.
>
> ### Re to response Q3:
>
> I agree with the performance improvement. However, you said,
>
> > note that the theoretical safety guarantee of CBF-QP relies on the existence of a “perfect” model.
>
> The proposed method also requires a perfectly safe MPC controller, which is no less strict than the CBF QP. Moreover, even with such a controller, you cannot guarantee intermediate safety theoretically.
>
> ### Re to response to Q5:
>
> >  In the problems, the policy that optimizes return is within the safe region by design.
>
> This is key information and should be highlighted in the manuscript. Then the proposed methods require a special reward design that the optimal policy is always safe, which is another limitation.
>
> Overall, and after reading reviewer zn6h’s concern about convergence, I will not change my score. I also encourage the authors to try to implement it on a real system if you do believe this is a reasonable way to do real-world safe exploration. The results will be much more powerful.

---

> > ### Author Response · Authors · 2024-08-10
> >
> > We thank the reviewer for the response.
> >
> > ### Re Re to response to W1:
> >
> > We agree that the existence of a safe MPC is an important consideration for the application and we will highlight this in section 3.1 the Safety Regularizer (where we introduce the estimated mode). Although parameter uncertainty in our experiments reflects one of many possible reasons for the model uncertainty, it is commonly used in the literature to simulate model uncertainty that is encountered in applications; one example is [1].
> >
> > ### Re Re to response to W3:
> >
> > The reviewer is correct, and we acknowledge that the safety of the intermediate action is not formally guaranteed. However, we made an addition to the manuscript following the feedback from you and another reviewer (t4ae), which may be of interest for your assessment. We have analytically quantified that the upper bound on the return deviation of the combined policy from the regularizer (safe) policy is proportional to $(1-\beta)\Delta a$, assuming the environment is Lipschitz continuous. This provides some theoretical insight on the safety of the combined policy.
> >
> > ### Re Re to response to Q1 and Q2, Q4:
> >
> > We are glad that these comments are addressed and appreciate the reviewer’s acknowledgement.
> >
> > ### Re Re to response Q3:
> >
> > As the reviewer identified, theoretically, both MPC and CBF-QP rely on the existence of a “safe” model. However, the fact that RL-ACR improves performance through the MPC objective justifies the use of proposed method over CBF-QP. We hope that this clarifies our original response.
> >
> > ### Re Re to response to Q5:
> >
> > We thank the reviewer for the valuable comment, and we will highlight this point in the paper. Also, we would like to point out, even reward designs where higher rewards correspond to preferable actions (in terms of safety) is common in the safe-critical applications (e.g., [2]). (In regulation tasks, for example, states closer to the target correspond to higher rewards and simultaneously states further away from the target are less safe states; thus increasing rewards corresponds to improving safety.)
> >
> > ### Re “after reading reviewer zn6h’s concern about convergence…”:
> >
> > We would like to mention that our initial response to zn6h’s included a vague term and raised a concern about convergence, but we have clarified this soon after in a follow-up response and made minor revisions to the manuscript to prevent any confusion.
> >
> > $~$
> >
> > [1] Cheng R, Orosz G, Murray RM, Burdick JW. End-to-end safe reinforcement learning through barrier functions for safety-critical continuous control tasks. InProceedings of the AAAI conference on artificial intelligence 2019 Jul 17 (Vol. 33, No. 01, pp. 3387-3395).
> >
> > [2] Laroche R, Trichelair P, Des Combes RT. Safe policy improvement with baseline bootstrapping. InInternational conference on machine learning 2019 May 24 (pp. 3652-3661). PMLR.

---

### Official Review · Reviewer_ta4e · 2024-07-05

**Soundness:** 3
**Presentation:** 4
**Contribution:** 3
**Rating:** 7
**Confidence:** 4

**Summary:**

In this article, the authors address the problem of safe reinforcement learning in single life setting. The agent must learn an optimal policy in one single episode without being unsafe throughout the learning. The proposed method relies on mixing a prior safe policy (the safety regularizer) with a reinforcement learning policy. The mixing is done through a linear combination of the actions proposed by the safety regularizer and the reinforcement learning agent. The weight put on either action is determined by a state dependent learned neural network.

The safety regularizer is given by a model predictive control algorithm which assumes an estimated model of the environment is available. The RL agent follows the SAC algorithm.

The authors show that the beta parameter is equivalent to a regularization weight that forces the learned policy to be closed to the regularizer policy.

They evaluate their algorithm empirically on several simulated environments that have mathematical models. The proposed method is able to outperform the baselines in terms of both safety and return.

I acknowledge reading the authors rebuttal.

**Strengths:**

The method is fairly easy to implement and is practical (under the condition that a safety regularizer exists). The overall idea is sound. The idea of mixing policies has been explored but they propose a principled way to learn the weights to give to both the safe policy and the learning policy which I think is useful. The idea of using MPC as a safe policy is original and interesting.

The author provide lemma and theorem that help with the interpretation of the method. They show that the mixing parameter they introduce can be interpreted as a regularization weight. They also show that the policy should converge under the optimal solution as beta goes to 1.

The method clearly outperforms the baselines in terms of safety level and provides better or equivalent performance in terms of return.

The article is overall well written and easy to follow.

**Weaknesses:**

- It would have been useful to quantify the deviation of the current policy from the safety regularizer as a function of $\beta$. Since the method does not provide any guarantee on the safety level, it would be useful to quantify the amount of risk being taken by not following strictly the safe policy.

- In the evaluation of the method, it would have been useful to add another baseline that takes advantage of the existence of a safe policy E.g. SPIBB (https://arxiv.org/abs/1712.06924), or residual policy learning (https://arxiv.org/abs/1812.06298). This last one would be a great baseline since they have a similar policy mixing concept but they don’t use any weight beta.
I acknowledge that the author have pre-trained SAC and CPO on the estimated model which is a good baseline as well.

- The framework is targeted towards “single life” settings, yet the training is episodic (figure 2). It makes it hard to conclude that the experiment are really evaluating the adaptability of the algorithm.

- Some of the results lack discussion. In particular, the authors claim that RL-ACR has less variance in the returns, it is not very obvious from figure 2 in my opinion. The figure clearly show that SAC is less stable but it is not really discussed why. Since RL-ACR is using SAC at its core, I would expect SAC without the regularizer to perform at least as well in terms of returns (with the caveat of more failures).

- The results shown in figure 3 are hard to evaluate as their lack some information about how the experiment is carried. Is it the performance in a single episode? Is it the same episode used for training?

- In figure 5, the authors claim “$\beta_{\psi}(s)$ can apply different policy combination weights depending on how well the current state is exploited”. I agree that it is state dependent but there is no explicit term that takes into account the “exploitation” level of a state. The experiment is not showing that either I believe. $\beta$ is learned by maximizing the $Q$ function estimate. If a state has not been visited often its Q values might still be overestimated and lead to high $\beta$.

Presentation:

-	The title and name of the algorithm is a bit misleading, I thought that the method would use techniques from the field of adaptive control https://en.wikipedia.org/wiki/Adaptive_control,  there is no actual analogy in the paper.

**Questions:**

-	Isn’t equation (10) equivalent to an actor loss in actor critic method? It could be useful to discuss a parallel here.
-	When updating $\pi_\theta$, what would be the effect of using gradient ascent on equation (10) instead?
-	Why is the performance of SAC worse than ACR in terms of normalized return?

**Limitations:**

The method requires the existence of a safety regularizer. In my opinion it is a fairly reasonable limitations and the method can still be used in many problems.
The method does not provide any safety guarantee.

---

> ### Author Rebuttal · Authors · 2024-08-07
>
> **[Weakness 1]**: The reviewer pointed out that it is useful to quantify the deviation of the combined policy from the regularizer policy.
>
> **Response**: We thank the reviewer for the interesting suggestion. We analytically quantified that for state $s$, the upper bound on the return deviation of the combined policy from the regularizer policy is proportional to $(1-\beta)\Delta a$, assuming the environment is Lipschitz continuous. The theorem and the proof sketch are given at the end of the rebuttal, and we will add the full proof to the paper.
>
> **[W2]**: The reviewer suggested adding another safe RL baseline.
>
> **Response**: We thank the reviewer for pointing out RPL as a baseline. RPL combines a safe policy with a learned residual policy and prioritizes the safe policy by initializing the residual policy as 0. However, the magnitude of the residual policy action can increase drastically. We have added RPL as an additional baseline, and our results show that RL-ACR is safer and more stable than RPL (see Figure R.I in the global rebuttal PDF).
>
> **[W3]**: The reviewer was concerned that the episodic training undermines the evaluation of the algorithms for the “single life” setting.
>
> **Response**: “Single life” describes the setting where no failure is tolerated during learning. Episodic training can be roughly regarded as introducing a disturbance every M step. Taking the Glucose problem, every episode is analogous to prescribing insulin dosages to a patient with off-balanced Glucose levels, and the single-life setting requires the algorithm not to fail in any (training or deployment) episode. Considering that RL-ACR instantly achieves the tasks, introducing further disturbances through episodic training adds difficulty to better evaluate its performance (as well as the baselines).
>
> **[W4]**: The reviewer raised that the results in Fig. 2 require more discussion as to why SAC shows larger return variances compared to RL-ACR.
>
> **Response**: We now add the discussion, which is covered in detail in the answer to **Q3**.
>
> **[W5]**: The reviewer pointed out that the experimental setting in Fig. 3 requires further explanation.
>
> **Response**: Fig. 3 shows the best performance after sufficient training, with methods deployed without exploration. We aim to show In Fig. 3: 1) how methods compare at their best, 2) RL-ACR improves the initial MPC policy, and 3) RL-ACR converges to a similar or stronger performance than standard SAC that disregards safety during training. We appreciate this comment and will clarify the above in the manuscript.
>
> **[W6]**: The reviewer questioned the effect of exploitation level on $\beta(s)$.
>
> **Response**: The $\beta(s)$ is updated iteratively using gradient ascent, thus the change in regularization level depends on the number of visits for state $s$ (besides the gradient). For any state $s$, the number of visits, which is the number of gradient ascent-based updates, affects the level to which the regularization level can be changed and thus reflects the exploitation level of state $s$.
>
> **[W7]**: The reviewer raised that the name of the algorithm might be confusing.
>
> **Response**: In the term “adaptive control regularization”, the “adaptive” describes “control-regularization”. However, we see the reviewer’s point that readers may think of “adaptive control”. We will change the term to “adaptive regularization”, to ensure clarity.
>
> **[Question 1]**: Discuss the similarity of Eq. (10) to the actor loss.
>
> **Answer**: We thank the reviewer for the comment and will add the following discussion: "Eq. (10) is similar to the actor loss in actor-critic methods. However, instead of updating the policy network, Eq. (10) updates $\beta(s)$ to find the optimal combination between the safe policy regularizer and the learned policy."
>
> **[Q2]**: What is the effect of using gradient ascent on Eq. (10) when updating $\pi_\theta$?
>
> **Answer**: Compared to Eq. (10), the RL-ACR loss has an additional entropy regularization term, which encourages exploring diverse actions and avoids sticking to the sub-optimum. Empirically, we tested the effect of using Eq. (10) to update $\pi_\theta$ (Figure R.III in the global rebuttal PDF), which showed reduced training stability.
>
> **[Q3]**: Explain the worse performance of SAC in Fig. 2.
>
> **Answer**: The difference between RL-ACR and SAC’s return variance comes from the regularizer in RL-ACR and its ability to prevent failure. Failure corresponds to a penalty affecting the return. SAC fails in some episodes (see the bottom panel of Fig. 2) and gets penalized, thus showing unstable returns compared to RL-ACR. As Fig. 3 shows, RL-ACR and SAC eventually converge to a similar return after sufficient training (as the reviewer identified). We will clarify the relationship between failure and observed return in Fig. 2.
>
> $~$
>
> **Theorem**: (Deviation from the Regularizer) Assume $R(s, a)$ and $P(s'|s, a)$ are $L_R$- and $L_P$-Lipshitz continuous. For $\forall s \in \mathcal{S}$, the following holds for the regularizer policy $\pi_r(s)$ and the combined policy $\pi_c(s)= \beta \pi_r(s) + (1-\beta(s)) \pi_\theta(s)$:
>
> $|V^{\pi_r}(s) - V^{\pi_c}(s)| \leq \frac{(1-\gamma)|\mathcal{S}|L_R + \gamma |\mathcal{S}| L_P R_\text{max}}{(1-\gamma)^2} (1-\beta(s)) \Delta a$
>
> **Proof Sketch**: Lipshitz continuous means:
> \begin{align}
>             &|R(s, \pi_r(s)) - R(s,\pi_c(s_t))| \leq L_R (1-\beta(s)) \Delta a \\\\
>             &\|P(\cdot|s, \pi_r(s)) - P(\cdot|s, \pi_c(s))\|_r \leq L_P (1-\beta(s)) \Delta a
> \end{align}
>
> According to the vectorized Bellman equation:
>
> $v_r-v_c = (I - \gamma P_r)^{-1}(r_r - r_c + \gamma (P_r - P_c) v_c)$
>
> Let $d_{., s}^T$ be the $s$-th row of $(I-\gamma P_.)^{-1}$  with entries $\leq 1/(1-\gamma)$. For $\forall s$:
>
> $|V_r(s) - V_c(s)| \leq |d_{r, s}^T (r_r - r_c)|+ \gamma| d_{c, s}^T (P_r -P_c)v_c|$.
>
> The proof follows by applying Holder's inequality to the two parts.

---

> > ### Comment · Reviewer_ta4e · 2024-08-08
> > **Thank you for the replies and for adding the RPL evaluation**
> >
> > The replies to the questions were very clear. The experiment with RPL was useful to convince me of the advantage of the method.
> >
> > Regarding weakness 6, I understand that the more frequent states will have more effect on the value of beta as you will do more gradient updates that will have these states in the batch, but I don't think we can really get insights on what beta will be for those states. Did I miss something?

---

> > > ### Author Response · Authors · 2024-08-09
> > >
> > > We thank the reviewer for the constructive feedback and for acknowledging our addressal of the comments. Regarding the new question, the reviewer is right, and we clarify this in the discussion of Fig. 5 that “exploitation is not explicitly taken into account by $\beta_\psi(s)$ but more frequent states will have more effect on the value of beta, although the exact value of beta cannot be predicted”.

---

### Official Review · Reviewer_ChN3 · 2024-07-11

**Soundness:** 3
**Presentation:** 3
**Contribution:** 2
**Rating:** 5
**Confidence:** 4

**Summary:**

The paper proposed RL-ACR algorithm to solve safe RL exploration problem where the RL policy must be safe during and after training. It achieves this by learning a "focus network" which mixes the action from MPC-based safety regularizer and conventional RL module. The algorithm is claimed to address "single-life" safe RL applications where safety is critical even during training.

**Strengths:**

1. Quality
* The approach itself is sound and the paper provided proof that the combined policy (which mixes actions from two modules) is unbiased if the RL module converges to optimal policy.
2. Clarity
* The paper is written in easy-to-understand manner and clearly illustrates how different components work together. Figures, equations and algorithm are all well-designed and well-written to support the entire paper.

**Weaknesses:**

1. Originality
* Most of the components proposed in RL-ACR have been explored previously: (i) combining two-prong policy [1], (ii) MPC with learned model [2, 3], (iii) safe RL with model rollout [4, 5, 6].
* The proposed focus network learns to outputs a scalar which is a mixing coefficient between the two policies. The focus network is similar to the learnable Safe Editor component in [1]. However, Safe Editor seems more sophisticated as it learns to edit the action itself. I wonder if there could be comparison between the two for evaluation.
2. Significance
* RL-ACR relies heavily on the the estimated model $\tilde{f}$ for safety. Consequently, the reliability (and accuracy) of the model is an important factor to guarantee safety. As pointed out in [4], even for a highly accurate model, it becomes increasingly challenging to predict future trajectory with reasonably good accuracy when the model unrolling horizon becomes larger because the error cascades over many timesteps. In Eq7 of the paper, RL-ACR is required to perform N step model unrolling. I have some doubts on how to obtain a reliable model to support RL-ACR.
* Related to previous point, in the conducted experiments, the model $\tilde{f}$ used in the paper is synthetically generated. It is not clear whether a model, which can predict lengthy future trajectory reliably, can be trained from ground-up. The tasks experimented (Glucouse, BiGlucouse, CSTR, Cart Pole) also do not seem to have long horizon or high dimension. Thus, I have reservation whether RL-ACR can be used to reliably solve safe exploration problems in more sophisticated tasks.
3. Baselines used in Experiments
* The safe RL baselines used in experiments: SAC & CPO are neither model-based nor intended for safe exploration. SAC only learns to maximize reward and disregards safety. It is also unclear how SAC makes use of the model in SAC-pt. For CPO, although it is a safe RL algorithm, it is not intended to solve safe exploration problem. It is also not clear to me how CPO-pt exploits the model $\tilde{f}$ for this purpose.
* Perhaps one possible baseline could be the safe editor in [1], which also claims to solve safe exploration problem?

References
[1] Yu, H., Xu, W. and Zhang, H., 2022. Towards safe reinforcement learning with a safety editor policy. Advances in Neural Information Processing Systems, 35, pp.2608-2621.
[2] Nagabandi, A., Kahn, G., Fearing, R.S. and Levine, S., 2018, May. Neural network dynamics for model-based deep reinforcement learning with model-free fine-tuning. In 2018 IEEE international conference on robotics and automation (ICRA) (pp. 7559-7566). IEEE.
[3] Chua, K., Calandra, R., McAllister, R., and Levine, S. Deep reinforcement learning in a handful of trials using probabilistic dynamics models. In Advances in Neural Information Processing Systems. 2018.
[4] Janner, M., Fu, J., Zhang, M. and Levine, S., 2019. When to trust your model: Model-based policy optimization. Advances in neural information processing systems, 32.
[5] Clavera, I., Fu, Y. and Abbeel, P., Model-Augmented Actor-Critic: Backpropagating through Paths. In International Conference on Learning Representations.
[6] Thomas, G., Luo, Y. and Ma, T., 2021. Safe reinforcement learning by imagining the near future. Advances in Neural Information Processing Systems, 34, pp.13859-13869.

**Questions:**

1. (Related to Weakness 3) Do you think safe editor policy in [1] can be a good a baseline for comparison?
2. (Related to Weakness 3) Do you have results which evaluate your approach in longer horizon (and higher state-action dimension) setting where it is more challenging to predict future trajectory reliably?
3. In your experiments, how do SAC-pt and CPO-pt make use of the model $\tilde{f}$? To my understanding, they are both model-free algorithms and do not exploit any dynamic model.

**Limitations:**

1. The paper discusses one limitation which is having reasonably accurate model for control regularizer. It is pointed out in [4] that even for a model which can predict the next step reliably, it is very difficult to predict future N steps accurately since the error cascade over the prediction timesteps. The paper could possibly discuss this in detail.
2. Another point about learning the model is that it should be able to predict transition to unsafe states in order to support the safety regularizer. This inevitably means that there needs to be unsafe transitions being collected to train the model. How does this model collection fit into the entire safe exploration framework is an interesting area to discuss.

---

> ### Author Rebuttal · Authors · 2024-08-07
>
> **[Weakness 1]**: The reviewer stated that some components of the proposed method have been previously explored.
>
> **Response**: While our work contains components of the suggested references, the resulting algorithm has a completely new approach to safety-critical control; in general, we believe that the combination of previously explored components does not take away from the originality of our work. Nevertheless, there are several key differences between our work and the mentioned references: i) Ref. [1] combines two learned policies, but RL-ACR regulates one learned policy with an established safe policy to ensure safety from the beginning of training. ii) Refs. [2, 3] use the MPC mainly to improve the “sample efficiency” rather than achieving “safe control” which is our aim; iii) In [4, 5, 6] the rollouts based on learned models only avoid unsafe actions after convergence, unlike RL-ACR that is designed to avoid safety violations from the beginning of training.
>
> **[W2]**: The reviewer raised that generating an estimated model for the safe MPC module is challenging. The reviewer was also concerned about whether the problems used in our experiments were complex enough.
>
> **Response**: For an environment with critical safety requirements, an estimated model (e.g. derived from limited observed data points) is sufficient for the safe MPC module of RL-ACR to perform very well. (One can think of the estimated model as a fit to the observed data points). This is shown in our Glucose and BiGlucose environments, where the model parameters are derived using real data from ~10 patients [7, 8]; this is practical in most real applications with limited data.
>
> We chose the environments to showcase the effectiveness of the method in “safety-critical” applications rather than focusing on environment complexity. Nevertheless, the BiGlucose model is generally regarded as a hard regulation problem, with 12 internal states (11 of them unobservable), 2 actions with large delays, and nondifferentiable piecewise dynamics. We have also provided further evidence of the effectiveness of RL-ACR in Acrobot, which is highly nonlinear and underactuated (see Figure R.II in the global rebuttal PDF).
>
> **[W3]**: The reviewer raised questions on the rationale for the selection of baselines SAC and CPO, and how SAC-pt and CPO-pt make use of the estimated model.
>
> **Response**: While it is true that SAC and CPO do not explicitly use models, the rationale behind our choice is threefold: i) SAC was chosen to show the optimal performance without considering safety. RL-ACR achieving stronger performance than SAC in Fig. 3 indicates that RL-ACR is not trading performance for safety. ii) CPO is a commonly chosen safe RL baseline that explicitly handles safety constraints during exploration (through CMDP formulation and recovery updates). iii) SAC-pt and CPO-pt are pretrained on the estimated model $\tilde{f}$ (which is used as a simulator of the actual environment); note that this is elaborated in the manuscript, see Line 217. This setting benefits SAC-pt and CPO-pt by allowing them to exploit the information in the available estimated model.
>
> **[Question 1]**: The reviewer asked if SEditor [1] can be used as a baseline.
>
> **Answer**: We thank the reviewer for pointing out this more up-to-date baseline and we have included it in our comparisons (see Figure R.I in the global rebuttal PDF). Results show SEditor from [1] does not guarantee safety during training because safety violations need to be observed before learning the cost return. SEditor also performs worse during training in 3 out of the 4 environments, potentially restricted by the constraint calculated from suboptimum cost estimation.
>
> **[Q2]**: The reviewer asked for results in environments where predicting the future is more challenging.
>
> **Answer**: We tested RL-ACR in Acrobot, where predicting the future is more challenging because of underactuation and high nonlinearity. Figure R.II in the PDF attached to the global rebuttal shows that RL-ACR maintains safety as only the first action in MPC rollout is adopted. Also, as mentioned in the response to **W2**, The Glucose and BiGlucose models are generally regarded as a hard regulation problem. The results demonstrate that RL-ACR is robust even against >60% model parameter mismatch (Fig. 4).
>
>
>
> **[Q3]**: The reviewer asked how SAC-pt and CPO-pt make use of the estimated model.
>
> **Answer**: SAC and CPO are model-free, thus not safe during training. We additionally generate SAC-pt and CPO-pt. which are pretrained using the estimated model $\tilde{f}$ as the simulator. This makes them more competitive by skipping the initial random trial-and-error. This also equips CPO-pt with safety knowledge through the pretrained cost value function, thus delivering more competitive safety performance upon deployment.
>
> $~$
>
> [7] Zahedifar, Rasoul, and Ali Keymasi Khalaji. "Control of blood glucose induced by meals for type-1 diabetics using an adaptive backstepping algorithm." Scientific Reports 12.1 (2022): 12228.
>
> [8] Hovorka, Roman, et al. "Nonlinear model predictive control of glucose concentration in subjects with type 1 diabetes." Physiological measurement 25.4 (2004): 905.

---

> ### Comment · Reviewer_ChN3 · 2024-08-13
>
> I thank the authors for their succinct and to-the-point rebuttal.
>
> The comparison with SEditor (Yu et al., 2022) is helpful as it demonstrated that RL-ACR is more suitable for environment with critical safety requirement where an established safe policy is helpful to ensure that. Due to the new results, I'm willing to increase my final rating.
>
> UPDATE: Final rating updated.

---

### Author Rebuttal · Authors · 2024-08-07

## Global Rebuttal

We thank the reviewers for reviewing the paper and providing valuable feedback. We have provided point-by-point responses to the weaknesses and questions listed by the reviewers. Here, we list a few important notes on some of the comments (some leading to additional data/experiment) that may be of interest to others.

1. In the original paper, we deliberately included methods that are recent but “well-established” as baselines in our experiments. As suggested by the reviewers, we have added two additional, more up-to-date, baselines: (silver et al., 2018) and (Yu et al., 2022), compared to which RL-ACR shows stronger performance in both safety and performance (see Figure R.I in the attached PDF).

2. For experimental analyses, we chose safety-critical environments to showcase the important applications of our algorithm. This includes some challenging environments with nondifferentiable environment dynamics and large delays. We have now provided some additional evidence on the robustness of RL-ACR in a highly nonlinear and underactuated environment, the Acrobot (Figure R.II in the PDF).

3. As suggested by reviewer ta4e, we performed an additional ablation study on the effect of entropy regularization in the actor loss (Figure R.III in the PDF), which shows that entropy regularization encourages diverse policies, and avoids sticking to a suboptimum policy.

4. Although some of the technical components presented in our work are explored in the existing literature, our proposed algorithm is unique in its ability to avoid control failure from the first training episode while simultaneously converging to the performance standards of RL methods that disregard safety; this is crucial for many real-world applications and yet very rare in the literature.

---

### Decision · Program_Chairs · 2024-09-25

**Decision:**

Accept (poster)

**Comment:**

Recently, safe reinforcement learning has been intensively studied for its application in many domains. This paper develops a new approach for safe RL exploration by combining the RL policy with an adaptive control regularizer.
The proposed algorithm, RL-ACR, ensures safety throughout the training process by incorporating a safety regularizer that helps avoid unsafe states. RL-ACR has analytical guarantees on unbiased convergence to the optimal policy in well-explored states while prioritizing safety in less-explored states.
The combination of adaptive control and RL is an important way to achieve safe RL.

As suggested by reviewers, more representative baselines could be added to make it a solid paper and justify the value of the theoretical analysis.The AC reads the paper and discussions and agrees with the reviewers that originality and baselines could be improved. After the rebuttals, the major concerns raised by the reviewers were addressed in the authors' responses.